# Banp regulates DNA damage response and chromosome segregation during the cell cycle in zebrafish retina

Swathy Babu, Yuki Takeuchi, Ichiro Masai*

Developmental Neurobiology Unit, Okinawa Institute of Science and Technology Graduate University, Onna, Japan

**Abstract** Btg3-associated nuclear protein (Banp) was originally identified as a nuclear matrix-associated region (MAR)-binding protein and it functions as a tumor suppressor. At the molecular level, Banp regulates transcription of metabolic genes via a CGCG-containing motif called the Banp motif. However, its physiological roles in embryonic development are unknown. Here, we report that Banp is indispensable for the DNA damage response and chromosome segregation during mitosis. Zebrafish *banp* mutants show mitotic cell accumulation and apoptosis in developing retina. We found that DNA replication stress and tp53-dependent DNA damage responses were activated to induce apoptosis in *banp* mutants, suggesting that Banp is required for regulation of DNA replication and DNA damage repair. Furthermore, consistent with mitotic cell accumulation, chromosome segregation was not smoothly processed from prometaphase to anaphase in *banp* morphants, leading to a prolonged M-phase. Our RNA- and ATAC-sequencing identified 31 candidates for direct Banp target genes that carry the Banp motif. Interestingly, a DNA replication fork regulator, *wrnip1*, and two chromosome segregation regulators, *cenpt* and *ncapg*, are included in this list. Thus, Banp directly regulates transcription of *wrnip1* for recovery from DNA replication stress, and *cenpt* and *ncapg* for chromosome segregation during mitosis. Our findings provide the first in vivo evidence that Banp is required for cell-cycle progression and cell survival by regulating DNA damage responses and chromosome segregation during mitosis.

*For correspondence:
masai@oist.jp

**Competing interest:** The authors declare that no competing interests exist.

## Editor's evaluation

A considerable knowledge gap exists in the understanding of BANP gene roles in vivo due to its essentiality, causing embryonic lethality in mice and humans. The authors here address this by exploiting the retinal cell proliferation and neurogenesis in zebrafish and define Banp roles in replication stress responses and mitosis involving p53, ATM and ATR signaling.

## Introduction

BTG3-associated nuclear protein (BANP) (also known as Scaffold/matrix attachment region binding protein 1 (SMAR1)) was identified as a nuclear protein that binds to the matrix-associated region (MAR) of the β-locus of murine T cell receptors (*Birot et al., 2000*; *Chattopadhyay et al., 2000*). MARs are often found near gene promoters (*Scheuermann and Garrard, 1999*) and constitute a periodic loop domain, through which chromosomal DNA is topologically organized inside the nucleus (*Blasquez et al., 1989*; *Hart and Laemmli, 1998*). Furthermore, BANP functions as a tumor suppressor (*Kaul Ghanekar et al., 2003*; *Kaul Ghanekar et al., 2005*). In humans, the *BANP* gene is located on chromosome 16q24, the genomic region of which often suffers from loss-of-heterozygosity in cervical and breast cancers (*Tsuda et al., 1994*; *Wang et al., 1999*). BANP interacts with tumor suppressor tp53 to

**eLife digest** In order for a cell to divide, it must progress through a series of carefully controlled steps known as the cell cycle. First, the cell replicates its DNA and both copies get segregated to opposite ends. The cell then splits into two and each new cell receives a copy of the duplicated genetic material. If any of the stages in the cell cycle become disrupted or mis-regulated this can lead to uncontrolled divisions that may result in cancer.

Researchers have often used a structure within the eye known as the retina to study the cell cycle in zebrafish and other animals as cells in the retina rapidly divide in a highly controlled manner. A protein called Banp is known to help stop tumors from growing in humans and mice, but its normal role in the body, particularly the cell cycle, has remained unclear. To investigate, Babu et al. studied the retina of mutant zebrafish that were unable to make the Banp protein.

The experiments revealed that two stress responses indicating DNA damage or defects in copying DNA were active in the retinal cells of the mutant zebrafish. This suggested that Banp allows cell to progress through the cell cycle by repairing any DNA damage that may arise during replication. Banp does this by activating the gene for another protein called Wrnip1. Babu et al. also found that Banp helps segregate the two copies of DNA during cell division by promoting the activation of two other proteins called Cenpt and Ncapg. Further experiments identified 31 genes that were directly regulated by Banp.

These findings demonstrate that Banp is required for zebrafish cells to be able to accurately copy their DNA and divide in to two new cells. In the future, the work of Babu et al. will provide a useful resource to investigate how tumors grow and spread around the body, and may contribute to the development of new treatments for cancer.

modulate its functions (*Kaul Ghanekar et al., 2003*; *Pavithra et al., 2009*; *Sinha et al., 2012*; *Sinha et al., 2010*). Overexpression of BANP suppresses cancer growth (*Bhagat et al., 2018*; *Liu et al., 2014*; *Taye et al., 2018*). However, these proposed cellular functions of BANP were based mostly on in vitro studies, and genetic deletion of *Banp* was reportedly unsuccessful (*Grand et al., 2021*), whereas *Banp* knockout mice show embryonic lethality (*Chemmannur et al., 2015*), which makes in vivo analysis difficult. Thus, physiological functions of BANP, especially in cell-cycle regulation and tumor suppression, are still unknown. Very recently, it was reported that BANP binds to the nucleotide sequence TCTCGCGAGA, called the 'Banp motif', which is enriched near the transcription initiation site of CpG island promoters, a genomic region in which it is difficult to replicate and repair DNA, and which regulates gene transcription in a DNA methylation-dependent manner. BANP promotes transcription of essential metabolic genes in pluripotent stem cells and terminal differentiated neuronal cells in humans and mice (*Grand et al., 2021*), indicating that BANP functions as a transcription factor at the molecular level. These findings raise the question, "How is the BANP-mediated transcriptional network involved in cell-cycle regulation and tumor suppression?".

The vertebrate retina is composed of six major classes of neurons and one type of glia (retinal ganglion cells [RGCs], three types of interneurons [amacrine cells, bipolar cells, and horizontal cells], two types of photoreceptors [rods and cones], and Müller glia). These retinal cells are assembled to form the neural circuit responsible for phototransduction and visual processing (*Dowling, 2012*). During development, retinal progenitor cells are multipotent and give rise to all seven retinal cell types in response to extrinsic and intrinsic cues (*Boije et al., 2014*). An important characteristic of retinal development is that retinal neurogenesis is linked to cell-cycle regulation. In zebrafish, cell-cycle length of retinal progenitor cells is around 32–49 hr from 6 to 25 hpf, but shortened to 8–10 hr once retinal neurogenesis starts at 25 hpf (*Li et al., 2000*), suggesting that activation of cell-cycle progression is associated with neurogenesis. It is generally accepted that active cell proliferation may increase DNA replication errors, which induce DNA replication stress and subsequent DNA damage (*Berti et al., 2020*; *Blackford and Jackson, 2017*; *Waterman et al., 2020*). Thus, factors involved in DNA replication checkpoint and DNA damage response are essential to ensure genomic integrity (*Murga et al., 2009*). Otherwise, retinal progenitor cells accumulate genetic defects leading to apoptosis. Consistently, in zebrafish, retinal apoptosis occurs in the absence of DNA damage response regulators such as DNA damage-binding protein 1 (DDB1) (*Yildirm et al., 2015*), Fanconi anemia

group D2 (Fancd2) (*Liu et al., 2003*), and Ku80 (Xrcc80) (*Bladen et al., 2005*). We also showed that retinal apoptosis occurs in zebrafish *prim1* mutants, in which DNA replication stress is chronically elevated. In this mutant, a central regulator of the DNA replication checkpoint, Ataxia telangiectasia-mutated and Rad3-related (ATR), is activated, which subsequently activates DNA damage response regulators, Ataxia telangiectasia-mutated (ATM) and Chk2, leading to tp53-mediated apoptosis. Thus, DNA replication checkpoint and DNA damage response function as safeguards to ensure integrity of retinal cell proliferation and neurogenesis (*Yamaguchi et al., 2008*). Therefore, developing retina is a highly proliferating tissue, in which a spatiotemporal pattern of neurogenesis is tightly coordinated by cell-cycle regulation. So, vertebrate retina provides a great model for studying how cell-cycle regulation, including DNA damage response, ensures neurogenesis and subsequent cell differentiation.

In this study, we identified zebrafish *banp* mutants, which show mitotic cell accumulation and apoptosis in the developing retina. DNA replication checkpoint and tp53-dependent DNA damage response are activated to induce apoptosis in *banp* mutant retinas. Thus, Banp is required for regulation of DNA replication and DNA damage repair. Furthermore, consistent with mitotic cell accumulation in *banp* mutants, our live imaging revealed that chromosome segregation fails to proceed smoothly during mitosis in *banp* morphants, resulting in a prolonged M-phase. Our RNA-sequencing and ATAC-sequencing enabled us to identify 31 candidates for direct Banp target genes that carry the Banp motif near their promoters. Interestingly, both zebrafish and human genes encoding two chromosome segregation regulators, *Cenpt* and *Ncapg*, and a DNA replication fork regulator, *Werner helicase interacting protein 1* (*Wrnip1*) are included in this list. Thus, Banp regulates transcription of *cenpt* and *ncapg* to promote chromosome segregation during mitosis, whereas Banp maintains transcription of *wrnip1* for recovery from DNA replication stress during S phase. Taken together, these data suggest that Banp is required for cell-cycle progression and cell survival in zebrafish retinas by regulating DNA damage response and chromosome segregation during mitosis.

## Results

### Zebrafish *rw337* mutants show mitotic cell accumulation and cell death in the retina after 2.5 dpf

We performed large-scale mutagenesis using zebrafish and identified a zebrafish mutant, *rw337*. *rw337* mutants have smaller eyes and a smaller optic tectum at 4 days post-fertilization (dpf) (*Figure 1A*). Around 6–7 dpf, *rw337* mutants die, indicating that this mutation is lethal. Next, we examined retinal phenotypes (*Figure 1B*). At 1.5 dpf, the retina consists of retinal progenitor cells and early differentiating neurons. There was no difference in retinal morphology between *rw337* mutants and wild-type siblings. However, at 2 dpf, mitotic-like round cells had accumulated in the apical region of the retina in *rw337* mutants, suggesting mitotic arrest or prolonged duration of mitosis. At 2.5 dpf, in wild-type retinas, three nuclear layers, the RGC layer, the inner nuclear layer (INL) and the outer nuclear layer (ONL), and two plexiform layers, the inner plexiform layer (IPL) and the outer plexiform layer (OPL), had formed. In *rw337* mutants, only the RGC layer was observed; however, many pyknotic nuclei were observed in the outer retinal region. At 3 dpf, the RGC layer, IPL and INL appeared in *rw337* mutants. However, pyknotic nuclei were densely packed in the outer region of *rw337* mutant retinas. At 4 dpf, most dead cell debris had decreased in *rw337* mutants. The RGC layer, IPL, and INL were formed; however, no ONL was observed, despite very small, patchy photoreceptor-like cells. At 3 and 4 dpf, the ciliary marginal zone (CMZ), where retinal stem and progenitor cells are located, was observed in *rw337* mutants. Thus, RGCs differentiate, but cell death occurs after 2.5 dpf in *rw337* mutants.

Next, we examined neuronal differentiation in *rw337* mutants at 4 dpf. The transgene *Tg[ath5:EGFP]* drives EGFP expression strongly in RGCs and weakly in amacrine cells and photoreceptors (*Figure 1—figure supplement 1A*; *Masai et al., 2000*; *Masai et al., 2005*). Pax6 is strongly expressed in amacrine cells and weakly in RGCs (*Figure 1—figure supplement 1C*; *Macdonald and Wilson, 1997*). Prox1 is expressed in bipolar cells and horizontal cells (*Figure 1—figure supplement 1E*; *Jusuf and Harris, 2009*). zpr1 antibody labels double cone photoreceptors (*Figure 1—figure supplement 1G*; *Larison and Bremiller, 1990*). Anti-glutamine synthetase (GS) antibody labels Müller cells (*Figure 1—figure supplement 1I*; *Peterson et al., 2001*). We observed that all these markers are expressed in *rw337* mutants (*Figure 1—figure supplement 1B, D, F, H, J*). However, the numbers of strong Pax6+ cells and GS+ cells were decreased in *rw337* mutants (*Figure 1—figure supplement 1D and J*),

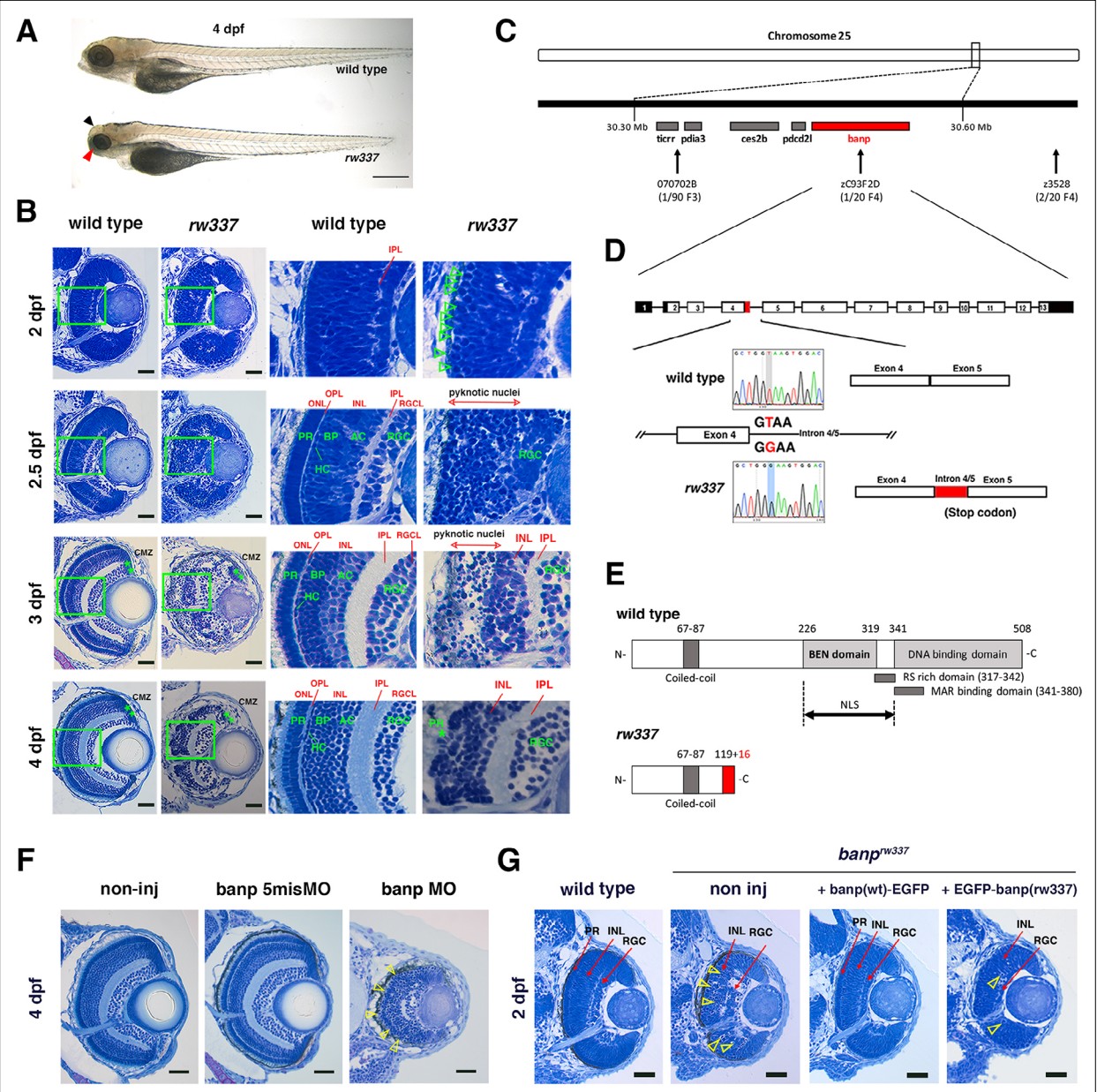

**Figure 1.** Zebrafish *banp^rw337* mutants show mitotic cell accumulation and cell death in the developing retina. (**A**) Morphology of wild-type and *rw337* mutant embryos at 4 dpf. *rw337* mutant embryos have smaller eyes compared to wild type at 4 dpf (red arrow). The black arrow shows the cloudy tectum in *rw337* mutant embryos, representing cell death. Scale bars: 100 μm. (**B**) Plastic sections of wild-type and *rw337* mutant retinas during development. Higher magnification images of green squares in the left columns are shown in the right columns. Retinal neuroepithelium is normal in *rw337* mutants until 2 dpf. However, many round cells reminiscent of mitotic cells are accumulated in the apical region of *rw337* mutant retinas at 2 dpf (open green arrowheads), suggesting mitotic defects in retinal progenitor cells. In wild-type retinas, three nuclear layers (RGCL, INL, and ONL) and two plexiform layers (IPL and OPL) are formed at 2.5 dpf. INL consists of amacrine cells (ACs), bipolar cells (BPs), and horizontal cells (HCs). ONL consists of two types of photoreceptors (PRs), rods and cones. However, only RGCs appear near the lens and the outer region was occupied by pyknotic nuclei, indicating dead cells in *rw337* mutants. At 3 dpf, INL is formed, but the outer region is still occupied by pyknotic nuclei in *rw337* mutants. At 4 dpf, pyknotic nuclei are reduced in *rw337* mutants; however, ONL and OPL are absent despite the small number of photoreceptor-like columnar cells in putative INL (PR, asterisks). At 3 and 4 dpf, the CMZ is maintained in *rw337* mutants (green double asterisks). (**C**) The *rw337* mutation is mapped in the genomic region flanked by two polymorphic markers, 070702B and zC93F2D, on chromosome 25. This genomic region contains four genes, including *banp*. (**D**) *banp* cDNA consists of 13 exons. In *rw337* mutants, T at the donor splice site of the intron between exons 4 and 5 (intron4/5) is converted to G, leading to the insertion of 52 nucleotides (red region) from intron 4/5. A new stop codon appears within this insertion. (**E**) Wild-type Banp protein has a coiled-coil domain, a BEN domain and a DNA-binding domain. An NLS domain, an RS-rich domain, and MAR-binding domain are shown. A truncated protein missing BEN and DNA-binding domains is predicted in *rw337* mutants. (**F**) Retinal sections of wild-type, wild-type embryos injected with banp-

*Figure 1 continued on next page*

*Figure 1 continued*

5misMO, and banp MO. *banp* morphant retinas have similar phenotypes to *banp^rw337^* mutants, with increased cell death and loss of the ONL. Yellow open arrowheads show pyknotic nuclei. (G) Retinal sections of wild-type retinas, *banp^rw337^* mutant retinas, and *banp^rw337^* mutant retinas overexpressing Banp(wt)-EGFP and EGFP-Banp(rw337). Overexpression of Banp(wt)-EGFP inhibits cell death and recovers retinal lamination in *banp^rw337^* mutants, whereas overexpression of EGFP-Banp(rw337) does not inhibit retinal defects in *banp^rw337^* mutants. Yellow open arrowheads indicate pyknotic cells in *banp^rw337^* mutant retinas and *banp^rw337^* mutant retinas overexpressing EGFP-Banp(rw337). Scale bars: 20 μm for (**B, F, G**).RGC, retinal ganglion cell; AC, amacrine cell, BP, bipolar cell; HC, horizontal cell; PR, photoreceptor; OPL, outer plexiform layer; IPL, inner plexiform layer; ONL, outer nuclear layer; INL, inner nuclear layer; RGCL, retinal ganglion cell layer; CMZ, ciliary marginal zone.

The online version of this article includes the following source data and figure supplement(s) for figure 1:

**Figure supplement 1.** Retinal cell differentiation in *banp^rw337^* mutants at 4 dpf.

**Figure supplement 2.** Amino acid conservation of Banp protein in zebrafish, mice and humans.

**Figure supplement 3.** *banp^sa12976^* homozygous mutants phenocopy *rw337* mutants.

**Figure supplement 4.** Confirmation of specificity of banp MO and nuclear localization of zebrafish Banp protein.

**Figure supplement 4—source data 1.** Data for *Figure 1—figure supplement 4D*.

**Figure supplement 5.** Overexpression of Banp(wt)-EGFP reduces cell death in *banp^rw337^* mutants.

**Figure supplement 5—source data 1.** Data for *Figure 1—figure supplement 5B*.

**Figure supplement 6.** Spatio-temporal pattern of zebrafish *banp* mRNA expression.

suggesting that amacrine cells and Müller cells were reduced. Prox1+ cells were misplaced and less numerous in *rw337* mutants, suggesting that horizontal and bipolar cells are also reduced (*Figure 1—figure supplement 1F*). zpr1+ cells were not detected, except for a small patch of zpr1+ cells in the ventral retina, where photoreceptor differentiation initially occurs in *rw337* mutants (*Figure 1—figure supplement 1H*), suggesting that most cone photoreceptors fail to differentiate. Anti-PCNA antibody labels retinal stem and progenitor cells in the CMZ (*Figure 1—figure supplement 1K*), which was not affected in *rw337* mutants (*Figure 1—figure supplement 1L*). Phalloidin labeling shows IPL and OPL in wild-type retinas (*Figure 1—figure supplement 1M*). *rw337* mutants showed IPL, but not OPL (*Figure 1—figure supplement 1N*), which is consistent with zpr1 labeling (*Figure 1—figure supplement 1H*). Thus, retinal stem and progenitor cells are maintained. RGCs do differentiate; however, amacrine cells, horizontal cells, bipolar cells, and Müller cells are decreased in number and photoreceptors fail to differentiate in *rw337* mutants.

## *rw337* mutant gene encodes Banp

We mapped the *rw337* mutation to zebrafish chromosome 25 (*Figure 1C*). Two polymorphic markers, 070702B and zC93F2D, restricted the genomic region encompassing *rw337* mutation, in which only four candidate genes: *BTG3-associated nuclear protein* (*banp*); *programmed cell death 2-like* (*pdcd2l*); *carboxylesterase 2b* (*ces2b*); *protein disulfide isomerase family A, member 3* (*pdia3*), were annotated. We cloned all these cDNAs from mRNA prepared from *rw337* homozygous mutants, sequenced them, and found that part of the intron sequence between exons 4 and 5 (intron 4/5) failed to be spliced out of the *banp* cDNA (*Figure 1D*). A nucleotide T at the exon-intron junction of exon 4 is converted to G, which alters the donor splice site of exon 4 from GT to GG. This mutation leads to abnormal insertion of a 52 bp intron sequence after exon 4, resulting in a premature stop codon in *banp* cDNA (*Figure 1D*). The *banp* gene encodes a protein containing a coiled-coil domain, a BEN domain, which was identified as a novel domain found in chromatin associated factors and DNA viral proteins and proposed to mediate protein-DNA and protein-protein interactions during chromatin organization and transcription (*Abhiman et al., 2008*), and a DNA-binding domain, which includes a MAR-binding domain (*Figure 1E*). Overall, zebrafish Banp protein shows high similarity to those of humans and mice, especially in the BEN domain (*Figure 1—figure supplement 2*). This mutation is predicted to generate nonfunctional, truncated Banp protein in *rw337* mutants (*Figure 1E*). There was no mutation in the other three genes in *rw337* mutants, suggesting that the *rw337* mutant gene encodes Banp.

To confirm that the loss of Banp causes retinal phenotypes in *rw337* mutants, we carried out three sets of experiments. First, we examined another nonsense mutation allele, *banp^sa12976^*, which has a premature stop codon in exon 4 (*Figure 1—figure supplement 3A*). Homozygous *banp^sa12976^* mutant embryos and trans-heterozygous *banp^rw337^; banp^sa12976^* mutants phenocopied *rw337* mutants (*Figure 1—figure supplement 3B*). Second, we inhibited Banp using ATG-morpholino (MO) antisense

oligos against Banp (banp MO). We confirmed MO specificity, because banp MO inhibited translation of mRNA encoding C-terminal EGFP-tagged wild-type Banp (banp(wt)-EGFP), whereas the control, 5-mismatch banp MO (banp 5misMO) did not (*Figure 1—figure supplement 4A and B*). *banp* morphants phenocopied *rw337* mutants (*Figure 1F*). Third, we overexpressed wild-type Banp in *banp^rw337* mutants. We injected mRNA encoding Banp(wt)-EGFP into wild-type embryos at the one-cell stage. In developing retinas, Banp(wt)-EGFP protein was localized in the nuclei during interphase, whereas it spread into the whole cell region during mitosis (*Figure 1—figure supplement 4C and D*), confirming nuclear localization of zebrafish Banp. We expressed Banp(wt)-EGFP, N-terminal EGFP tagged wild-type (EGFP-Banp(wt)) and the *rw337* mutant form of Banp (EGFP-Banp(rw337)) in wild-type retinas. Only EGFP-Banp(rw337) was not stably maintained (*Figure 1—figure supplement 4E*). Finally, overexpression of Banp(wt)-EGFP rescued retinal phenotypes in *rw337* mutants at 2 dpf, but overexpression of EGFP-Banp(rw337) did not (*Figure 1G* and *Figure 1—figure supplement 5*). Taken together, retinal phenotypes of *rw337* mutants are caused by the loss of Banp.

## *banp* mRNA is prominently expressed in the developing CNS of zebrafish

We examined *banp* mRNA expression by in situ hybridization (*Figure 1—figure supplement 6A*). *banp* mRNA shows ubiquitous expression from the one-cell stage to the 21-somite stage, suggesting maternal and zygotic expression. mRNA was restricted to the brain, including the retina at 30 and 48 hpf. Frontal sections show that mRNA is expressed in neural retina at 24 hpf, but restricted in the CMZ at 48 hpf, suggesting that retinal progenitor cells express *banp* mRNA (*Figure 1—figure supplement 6B*). In addition to brain and retina, *banp* mRNA expression was observed in fin and notochord at 48 hpf and in neuromasts of the lateral line after 72 hpf. *banp* mRNA expression was not detected in *banp^rw337* homozygous mutants at 3 dpf (*Figure 1—figure supplement 6C*), suggesting mRNA decay. Thus, *banp* mRNA is prominently expressed in developing brain in zebrafish.

## *banp* mutants show mitotic defects and cell death of retinal progenitor cells

Although retinal progenitor cells produce RGCs in *banp ^rw337* mutant retinas, the number of other neurons, especially photoreceptors, was decreased at 4 dpf (*Figure 1—figure supplement 1*). Indeed, mitotic-like round cells and pyknotic nuclei were accumulated in the outer region of *banp^rw337* mutant retinas from 2 to 3 dpf (*Figure 1B*), suggesting that mitotic defects and cell death reduce the number of retinal neurons. Thus, we conducted double labeling of wild-type and *banp^rw337* mutant retinas with TdT-mediated dUTP Nick End Labeling (TUNEL), which labels blunt ends of DNA double-strand breaks (DSBs) to visualize apoptosis, and with anti-phosphorylated (Ser-10) Histone H3 (pH3) antibody, which labels mitotic cells in prometaphase, metaphase and early anaphase (*Prigent and Dimitrov, 2003*; *Figure 2A*). Then, to evaluate TUNEL and pH3 signals more precisely, we used only fluorescent channel for TUNEL (*Figure 2B*) or anti-pH3 antibody labeling (*Figure 2B'*), with nuclear staining with Sytoxgreen. TUNEL+ cells were rarely observed in wild-type retinas throughout stages from 29 to 101 hpf, as well as in *banp^rw337* mutant retinas at 24 hpf (*Figure 2B*). However, in *banp^rw337* mutants, TUNEL+ cells appeared in the neural retina at 53 hpf, and became more localized to the outer retina at 77hpf (*Figure 2B*) and to the interface between the central retina and the CMZ at 77 and 101 hpf (*Figure 2B*). After 53 hpf, the number of TUNEL+ cells was significantly higher in *banp^rw337* mutants than in wild-type siblings (*Figure 2C*). Thus, retinal cells undergo apoptosis in *banp^rw337* mutants. Next, we examined mitosis by labeling with anti-pH3 antibody (*Figure 2B'*). The number of pH3+ cells did not differ between wild-type and *banp^rw337* mutant retinas at 24 hpf. However, pH3+ cells were markedly increased and located at the apical surface of the neural retina in *banp^rw337* mutant retinas at 53 hpf (*Figure 2B'*), confirming mitotic cell accumulation. At 77 and 101 hpf, pH3+ cells disappeared from the central retina, but still were accumulated in the CMZ in *banp^rw337* mutants (*Figure 2B'*). The number of pH3+ cells in *banp^rw337* mutant retinas increased transiently at 53 hpf, and then decreased, but was still significantly higher than that of wild-type sibling retinas after 77 hpf because of accumulation of pH3+ cells at the CMZ (*Figure 2D*).

To clarify how mitotic cell accumulation and apoptosis are induced during retinal cell proliferation and neurogenesis in *banp^rw337* mutants, we examined BrdU incorporation and ath5:EGFP expression at 48 hpf (*Figure 2—figure supplement 1A and B*). The percentage of BrdU+ area to total retinal

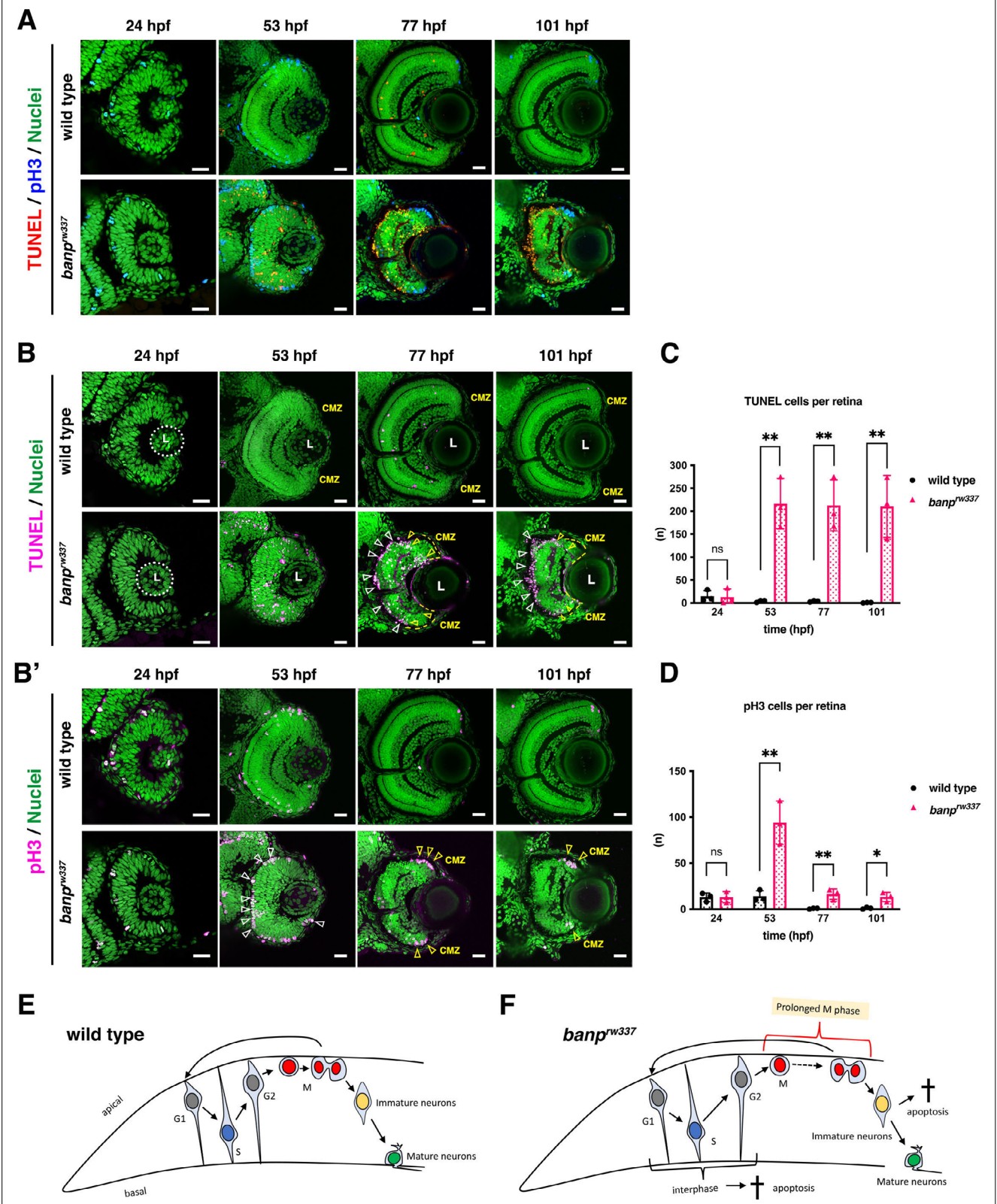

**Figure 2.** *banp* mutant retinas show mitotic cell accumulation and apoptosis. (**A**) Double labeling of wild-type (upper) and *banp^rw337* mutant (lower) retinas with TUNEL (red) and anti-pH3 antibody (blue). Nucleus was counterstained with Sytoxgreen (green). (**B**) Only TUNEL and nuclear fluorescent channel of images shown in (**A**). TUNEL and nuclear staining are indicated in magenta and green, respectively. In *banp^rw337* mutants, apoptotic cells start to be observed in neural retinas at 53 hpf, and increase in the apical region of retinas (white arrowheads) and at the interface between the CMZ and

*Figure 2 continued on next page*

*Figure 2 continued*

the central retina (yellow open arrowheads) at 77 and 101 hpf. (**B'**) Only pH3 and nuclear fluorescent channel of images shown in (**A**). pH3 signals and nuclear staining are indicated in magenta and green, respectively. Mitotic cells are accumulated at the apical surface of *banp*$^{rw337}$ mutant retinas at 53 hpf (white open arrowheads) and restricted in the apical region of the CMZ (yellow open arrow) at 77 and 101 hpf. (**C**) The number of TUNEL-positive cells per retina. Apoptosis is significantly increased in *banp*$^{rw337}$ mutants after 53 hpf. Unpaired t-test, two-tailed, Mean ± SD. [n=3, p**<0.01] (**D**) The number of pH3-positive cells per retina. The number of mitotic cells is significantly increased in *banp*$^{rw337}$ mutants at 53 hpf. The number is decreased, but still significantly higher in *banp*$^{rw337}$ mutants than in wild-type siblings after 77 hpf. Unpaired t-test, two-tailed, Mean ± SD. [n=3, p**<0.01] (**E**) Schematic diagram of wild-type retinas. Retinal progenitor cells undergo cell-cycle progression in which the nuclei move along the apico-basal axis depending on cell-cycle phase (the most basal location in S phase and the most apical location in M phase), called interkinetic nuclear migration. Mitosis occurs in the apical surface of the neural retina. Daughter cells reenter the cell cycle as retinal progenitor cells or start to differentiate into postmitotic neurons, which migrate into the basal region and become mature neurons. (**F**) Schematic diagram of *banp*$^{rw337}$ mutant retinas. A fraction of interphase progenitor cells (G1, S, or G2 phase cells) undergo apoptosis after 48 hpf. Mitotic cells are accumulated at the apical surface, which may be caused by prolonged mitotic duration. A fraction of immature differentiating neurons also undergo apoptosis. Thus, *banp*$^{rw337}$ mutant retinal cells show multiple defects in cell-cycle progression of retinal progenitor cells and retinal neurogenesis. Scale bars: 20 μm for (**A, B, B'**).

The online version of this article includes the following source data and figure supplement(s) for figure 2:

**Source data 1.** Data for *Figure 2CD*.

**Figure supplement 1.** Apoptosis is induced in retinal progenitor cells and newly differentiating neurons in *banp*$^{rw337}$ mutants.

**Figure supplement 1—source data 1.** Data for *Figure 2—figure supplement 1C, F, G, I, K*.

---

area did not differ significantly between wild-type siblings and *banp*$^{rw337}$ mutants (***Figure 2—figure supplement 1C***), whereas the percentage of ath5:EGFP+ area to the total retinal area was slightly, but significantly lower in *banp*$^{rw337}$ mutants than in wild-type siblings (***Figure 2—figure supplement 1C***), suggesting that retinal neurogenesis is slightly reduced in *banp*$^{rw337}$ mutants at 48 hpf, probably due to mitotic defects or apoptosis.

Next, we examined the spatial position of mitotic cells and apoptosis at 48 hpf by labeling with anti-pH3 and anti-activated caspase 3 antibodies (***Figure 2—figure supplement 1D and E***). Both caspase 3+ and pH3+ cells were increased in *banp*$^{rw337}$ mutants (***Figure 2—figure supplement 1F***). Furthermore, caspase 3+ cells were mostly located in the intermediate zone of the neural retina along the apico-basal axis, whereas pH3+ cells were localized at the apical surface of the neural retina (***Figure 2—figure supplement 1G***), suggesting that apoptosis occurs mostly in retinal progenitor cells undergoing G1, S, or G2 phase, or differentiating neurons. Next, we examined whether apoptosis occurs in early differentiating retinal neurons, using double labeling with anti-activated caspase 3 antibody and ath5:EGFP (***Figure 2—figure supplement 1H***). The percentage of ath5:EGFP- and caspase 3-double-positive cells to the total number of caspase 3+ cells was 9.44% ± 13.1% in *banp* mutants at 48 hpf and higher than that of wild-type siblings (0.00% ± 0.00%), although the difference was not significant (***Figure 2—figure supplement 1I***). Next, we examined whether apoptosis occurs in mature retinal neurons. HuC/D is expressed in maturing neurons (***Park et al., 2000***). At 48 hpf, anti-HuC/D antibody labeled RGCs in zebrafish (***Figure 2—figure supplement 1J***). The fraction of caspase 3- and HuC/D-double-positive cells among total caspase 3+ cells was almost zero in both wild-type and *banp*$^{rw337}$ mutants (***Figure 2—figure supplement 1K***), suggesting that apoptosis does not occur in maturing retinal neurons. Taken together, in *banp*$^{rw337}$ mutants, retinal progenitor cells primarily show mitotic defects at 48 hpf. On the other hand, apoptosis is likely induced in retinal progenitor cells at interphase or in newly differentiating retinal neurons, although apoptosis in newly differentiating retinal neurons is minor (***Figure 2—figure supplement 1L***). Since mitotic cell accumulation was transiently observed in the central retina at 53 hpf (***Figure 2B'***), and apoptosis was not overlapped with this mitotic cell accumulation area (***Figure 2—figure supplement 1G***), it is less likely that mitotic cells undergo permanent mitotic arrest leading to apoptosis. Rather, mitotic cell accumulation seems to be caused by prolonged mitosis. Thus, *banp*$^{rw337}$ mutants show mitotic cell accumulation and apoptosis in retinal progenitor cells during interphase and immature neurons (***Figure 2E and F***).

## tp53 pathway mediates retinal apoptosis in *banp* mutants

To understand the molecular mechanism of mitotic defects and apoptosis in *banp*$^{rw337}$ mutants, we identified differentially expressed mRNAs using RNA-seq analysis. We prepared mRNA from 48-hpf wild-type siblings and *banp*$^{rw337}$ mutants. Illumina sequencing generated more than 50 million paired-end reads from cDNA libraries. 27,771 protein-coding genes were detected. Among them,

258 genes were upregulated, and 81 genes were downregulated in *banp*[rw337] mutants (*Figure 3A*). The top 15 upregulated genes contained a significant fraction of tp53 dependent-DNA damage response regulators (*Figure 3B*). Quantitative RT-PCR revealed that four tp53 target genes: *p21(cdkn1a), puma (bbc3), ccng1*, and *mdm2* are significantly upregulated in *banp*[rw337] mutants (*Figure 3C*). Western blotting showed that full length tp53 (FL tp53) protein is stabilized and more than 11-fold higher in *banp*[rw337] mutants than in wild type (*Figure 3D and E*). We previously showed that the ATR/ATM-Chk2-tp53 DNA damage pathway induces retinal apoptosis in zebrafish (*Yamaguchi et al., 2008*). Indeed, the top 15 upregulated genes contain exo5, a critical regulator for ATR-dependent replication fork restart (*Hambarde et al., 2021*; *Figure 3B*). mRNA expression of ATR and ATM was upregulated in *banp*[rw337] mutants (*Figure 3F*). So, we examined whether retinal apoptosis depends on tp53 in *banp*[rw337] mutants. Injection of tp53 antisense morpholino (tp53 MO) completely inhibited retinal apoptosis in *banp*[rw337] mutants at 48 hpf (*Figure 3G and H*). However, retinal apoptosis reappeared in those tp53 MO-injected *banp*[rw337] mutants at 72 hpf (*Figure 3G*), although there was no statistical difference in apoptotic cell number between wild-type and *banp*[rw337] mutant retinas at 72 hpf (*Figure 3H*). Thus, retinal apoptosis in *banp*[rw337] mutants depends on tp53, although it is possible that some tp53-independent mechanism induces retinal apoptosis after 48 hpf.

## Transcription of Δ113-tp53 is preferentially elevated in *banp* mutants

RNA-seq analysis showed upregulation of *tp53* mRNA (*Figure 3B*); however, surprisingly, quantitative RT-PCR showed that the mRNA level of FL tp53 is not significantly different between *banp*[rw337] mutants and wild-type siblings at 48 hpf (*Figure 4A*). There are several isoforms of tp53 protein (*Joruiz et al., 2020*; *Joruiz and Bourdon, 2016*). Importantly, in zebrafish, the isoform Δ113tp53 is generated from the alternative transcription start site in intron 4. It was reported that FL tp53 initially activates transcription of Δ113tp53 after tp53 protein is stabilized in response to cellular stress (*Chen et al., 2009*). Δ113tp53 preferentially activates transcription of target genes related to cell-cycle arrest to promote DNA repair and cell survival, whereas FL tp53 preferentially activates transcription of target apoptotic genes to eliminate severely DNA-damaged cells (*Gong et al., 2015*; *Gong et al., 2020*). Differential expression of tp53 isoforms is thought to regulate the balance between cell survival and apoptosis in response to DNA damage. Indeed, quantitative RT-PCR confirmed that only Δ113tp53 was upregulated in *banp*[rw337] mutants (*Figure 4A*). ATAC-seq revealed that a genomic region in intron 4 of tp53 is specifically open in *banp*[rw337] mutants, but not in wild-type siblings (*Figure 4B*, ATAC-seq). Consistently, transcription of the Δ113tp53 mRNA was specifically activated in *banp*[rw337] mutants, but not in wild-type siblings (*Figure 4B*, RNA-seq). Thus, the internal promoter at intron 4 (*Chen et al., 2009*; *Chen et al., 2005*) is specifically activated in *banp*[rw337] mutants (*Figure 4B*). These data raise the possibility that the lack of Banp increases DNA replication stress and promotes an accumulation of DNA damage, resulting in activation of ATR and ATM, which stabilize FL tp53, such that FL tp53 triggers Δ113tp53 transcription in *banp*[rw337] mutants (*Figure 4G*).

## DNA damage accumulates during replication in *banp* mutant retinas

To examine whether replicative DNA damage is increased due to loss of Banp functions, we labeled 48-hpf retinas with anti-γ-H2AX antibody. In response to DSBs, ATM rapidly phosphorylates histone H2A isoform H2AX at a C-terminal serine-139, which is called γ-H2AX (*Blackford and Jackson, 2017*; *Rogakou et al., 1998*; *Shroff et al., 2004*). Furthermore, ATR phosphorylates H2AX at stalled replication forks, and later, when single-ended DSBs form at persistent stalled forks, ATM and DNA-PKs act jointly to further propagate γ-H2AX (*Sirbu et al., 2011*; *Ward and Chen, 2001*). Thus, γ-H2AX is a global marker of stalled replication forks and DSBs. The number of γ-H2AX+ cells increased in both *banp*[rw337] mutant (*Figure 4C and F*) and *banp* morphant retinas (*Figure 4—figure supplement 1A and B*), confirming an accumulation of DNA damage in the absence of Banp. Next, to determine how DNA damage is induced in *banp*[rw337] mutants, we examined the fraction of S-phase cells, mitotic cells, and post-mitotic neurons in γ-H2AX+ retinal cells in *banp*[rw337] mutants at 48 hpf. First, wild-type and *banp*[rw337] mutant retinas carrying the transgene *Tg[ath5:EGFP]* were labeled with anti-BrdU and anti-γ-H2AX antibodies (*Figure 4—figure supplement 1C*). The fraction of γ-H2AX+ cells increased in *banp*[rw337] mutant retinas, whereas the fraction was almost zero in wild-type sibling retinas (*Figure 4—figure supplement 1D*). Next, we compared the fraction of BrdU+ cells between the total retinal cells and γ-H2AX+ retinal cells in *banp*[rw337] mutants (*Figure 4—figure supplement 1E*). The BrdU+ fraction

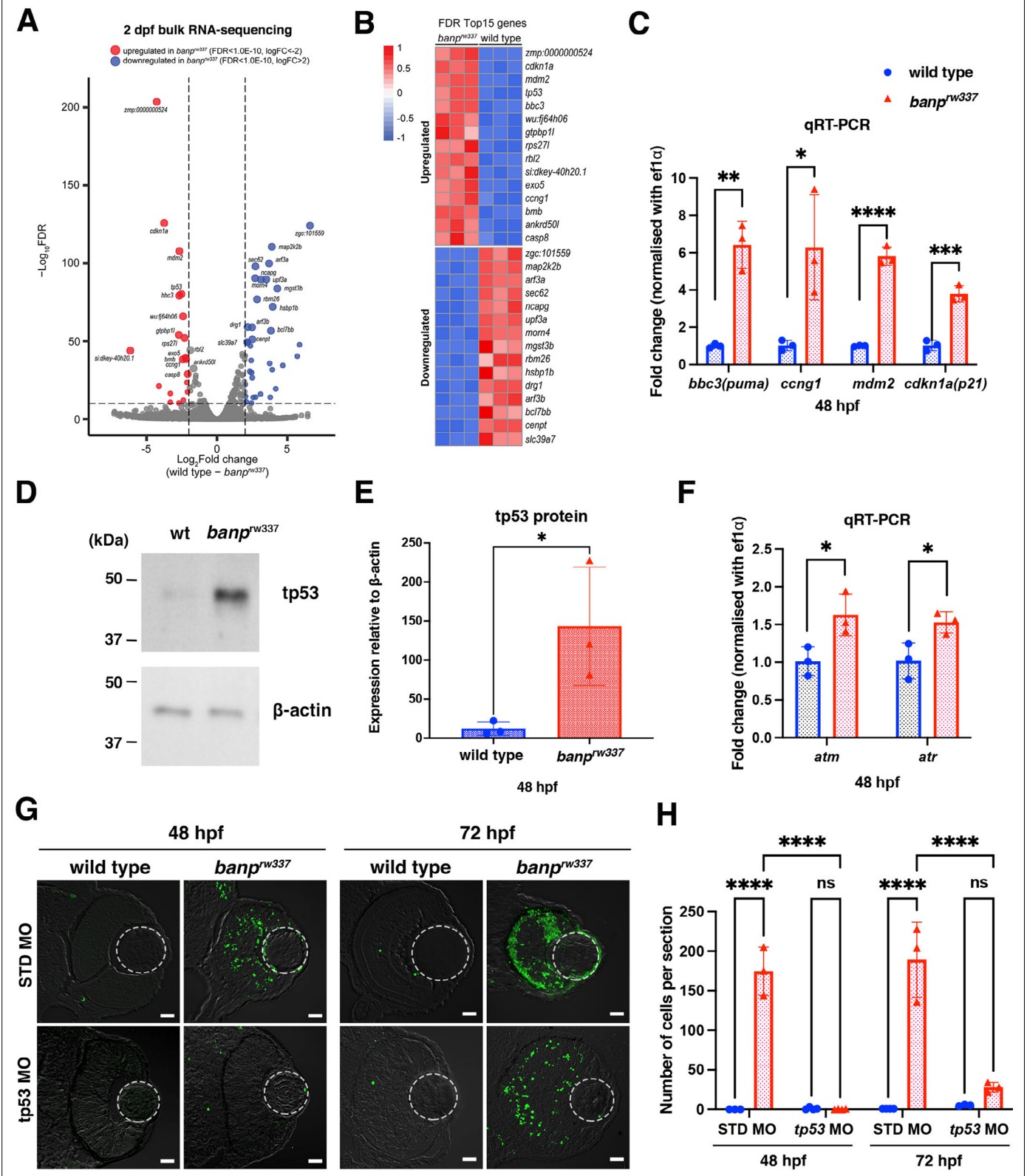

**Figure 3.** tp53-mediated DNA damage response is activated in the absence of Banp. (**A**) A volcano plot indicating gene differential expression of 48 hpf wild-type versus *banp^rw337* mutant embryos. Red dots show genes that were upregulated ≥2 fold in mutants, while blue dots represent genes downregulated ≥2 fold. (**B**) Heat map of the top 15 differentially expressed genes. (**C**) Relative mRNA expression levels for *bbc3*, *ccng1*, *mdm2*, *cdkn1a* in wild type and *banp^rw337* mutants. Quantitative RT-PCR confirmed the up-regulation of p53 pathway genes in *banp^rw337* mutants at 48 hpf. Unpaired

*Figure 3 continued on next page*

*Figure 3 continued*

t-test (two-tailed), Mean ± SD. [n=3, p*<0.05, p**<0.01, p***<0.001, p****<0.0001]. (**D**) Western blotting of wild-type and *banp^rw337* mutants with anti-tp53 antibody. Full-length tp53 protein is stabilized in *banp^rw337* mutants, compared to wild type. β-actin protein is an endogenous control and used for normalization of tp53 protein level. (**E**) Quantification of tp53 protein level relative to a control protein, β-actin, in wild-type and *banp^rw337* mutants. The tp53 protein level is significantly higher in *banp^rw337* mutants than in wild type at 48 hpf. Unpaired t-test (two-tailed), Mean ± SD. [n=3, p*<0.05]. (**F**) Relative mRNA expression levels for *atm* and *atr* in wild type and *banp^rw337* mutants. Both mRNAs are significantly up-regulated in *banp^rw337* mutants at 48 hpf. Unpaired t-test (two-tailed) [n=3, p*<0.05]. (**G**) TUNEL of wild-type and *banp^rw337* mutant retinas injected with STD MO and tp53 MO at 48 hpf and 72 hpf. Scale bars: 20 μm. (**H**) Histogram of the number of TUNEL-positive cells per retinal section in wild-type and *banp^rw337* mutant retinas injected with STD MO and tp53 MO at 48 hpf and 72 hpf. Two-way ANOVA, Tukey's multiple comparisons test, Mean ± SD. [n=3 for STD-MO at 48 hpf and tp53-MO at 72 hpf; n=4 for tp53-MO at 48 hpf and STD-MO at 72 hpf, p****<0.0001].

The online version of this article includes the following source data for figure 3:

**Source data 1.** Data for *Figure 3CEFH*.

**Source data 2.** Data for *Figure 3D*.

was significantly higher in γ-H2AX+ cells than in total retinal cells (*Figure 4—figure supplement 1F*), indicating that S-phase fraction is 4.3-fold higher in DNA damaged retinal cells than in total retinal cells, in *banp^rw337* mutants. Thus, replicative DNA damage is abnormally increased in retinal progenitor cells undergoing S phase. Next, we examined the fraction of ath5:EGFP+ cells (*Figure 4—figure supplement 1G*). There was no significant difference in the ath5:EGFP+ cell fraction between the total retinal cells and γ-H2AX+ retinal cells in *banp^rw337* mutants (*Figure 4—figure supplement 1H*). Finally, we determined the fraction of mitotic cells by labeling with anti-pH3 and anti-γ-H2AX antibodies in *banp^rw337* mutant retinas at 48 hpf (*Figure 4—figure supplement 1I*). There is no significant difference in the pH3+ cell fraction between the total retinal cells and γ-H2AX+ retinal cells (*Figure 4—figure supplement 1J*). Despite of no statistical significance, interestingly, averaged fraction of pH3+ cells in γ-H2AX+ retinal cells was lower than that of total retinal cells. Since *banp^rw337* mutants show increase in γ-H2AX+ cells, pH3+ cells and apoptotic cells at 48 hpf, it is likely that DNA damaged cells are eliminated by apoptosis in S phase prior to mitosis. This also suggests that M phase defects is independent of replicative DNA damage introduced in S phase. Although DNA damage may occur in post-mitotic neurons and mitotic cells in *banp^rw337* mutants, it is likely that most DNA damage is introduced in S-phase of retinal progenitor cells due to replication fork stalling in the absence of Banp.

It is generally accepted that tp53 promotes DNA damage repair or eliminates DNA-damaged cells via apoptosis, so we expected that tp53 knockdown might enhance accumulation of DNA damage in *banp^rw337* mutants. However, compared with standard-MO injection control (n=120.8 ± 60.6/section), the injection of FL tp53-MO significantly reduced the number of γ-H2AX+ cells in *banp^rw337* mutant retinas (n=55.9 ± 15.9/section) at 48 hpf (*Figure 4C and F*). Injection of Δ113tp53-MO mildly reduced the number of γ-H2AX+ cells (n=85.4 ± 15.2/section); however, the difference was not significant (*Figure 4C and F*). It is likely that neither tp53 nor Δ113tp53 repairs DNA damage in *banp^rw337* mutants. Thus, under normal conditions, Banp probably suppresses DNA replication stress upstream of tp53 by promoting recovery from stalled forks or repair of DSBs (*Figure 4G*).

We already showed that, compared with Standard-MO injection controls (n=174.7 ± 30.5/section), retinal apoptosis was completely inhibited by FL tp53-MO in *banp^rw337* mutant retinas (n=0.50 ± 0.58/section) at 48 hpf (*Figure 3G and H*; also shown in *Figure 4C and D*). Next, we investigated whether retinal apoptosis in *banp^rw337* mutants depend on Δ113tp53-MO, and compared inhibition levels of retinal apoptosis between FL tp53-MO and Δ113tp53-MO. However, inhibition of apoptosis in *banp^rw337* mutant retinas by Δ113tp53-MO was significant, but ineffective (n=58.00 ± 11.0/section) (*Figure 4C and D*). This result is consistent with previous reports proposing that Δ113tp53 preferentially activates transcription of target genes related to cell-cycle arrest rather than apoptosis (*Gong et al., 2015*; *Gong et al., 2020*).

Furthermore, we investigated whether mitotic cell accumulation in *banp^rw337* mutants depend on FL tp53-MO or Δ113tp53-MO. Compared with Standard-MO injection controls (n=376.6 ± 74.8/section), the number of pH3+ cells in *banp^rw337* mutants was significantly decreased by FL tp53-MO (n=262.4 ± 68.3/section) (*Figure 4C and E*). However, the number of pH3+ cells was still higher in *banp^rw337* mutant retinas injected with FL tp53-MO than in wild-type control retinas injected with FL tp53-MO, indicating that there is a substantial tp53-independent fraction. Furthermore, the injection of Δ113tp53-MO did not significantly rescue mitotic phenotypes in *banp^rw337* mutants (n=303.8 ± 46.6/

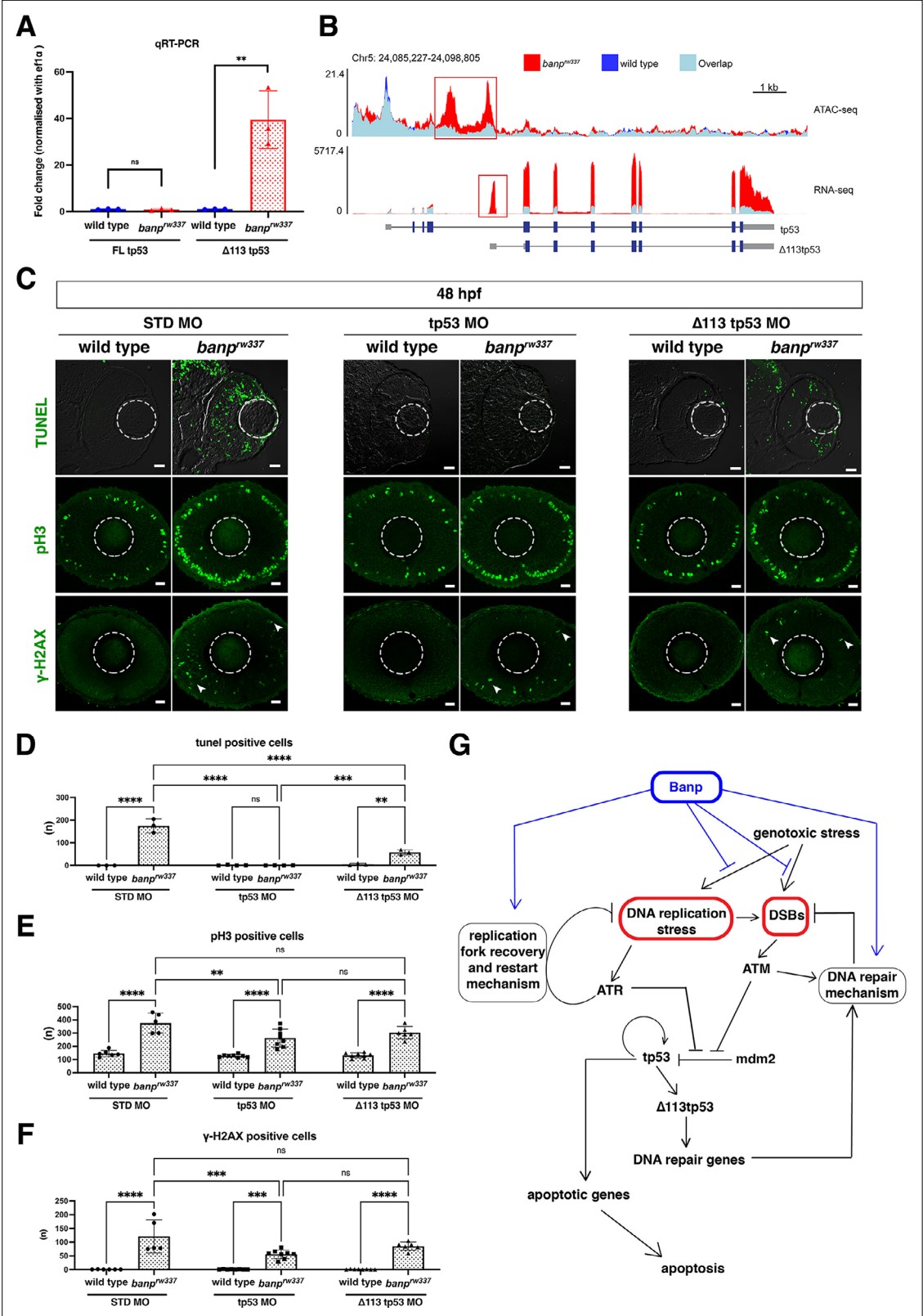

**Figure 4.** Banp is required for integrity of DNA replication and DNA damage repair. (**A**) Quantitative RT-PCR-based validation of FL tp53 and Δ113tp53 mRNA expression at 48 hpf. Δ113tp53 mRNA expression is markedly higher in *banp^rw337* mutants than in wild-type siblings, whereas FL tp53 mRNA is not significantly different between mutants and wild-type siblings. Unpaired t-test (two-tailed), Mean ± SD. [n=3, p**<0.01, ns (not significant)]. (**B**) ATAC-seq- and RNA-seq-based validation of chromatin accessibility (upper panel) and transcription level (bottom panel) of the genomic region covering

*Figure 4 continued on next page*

*Figure 4 continued*

both tp53 and Δ113tp53 transcription units. Data of wild-type and *banp^rw337* mutants are indicated in blue and red, respectively. Light blue indicates overlapping level. The red openbox in ATAC-seq shows open chromatin in *banp^rw337* mutants in intron 4, signifying differential transcription of Δ113tp53 using an internal promoter. (C) TUNEL (top), anti-pH3 antibody labeling (middle), and anti-γ-H2AX antibody labeling (bottom) of wild-type and *banp^rw337* mutant retinas injected with STD MO, tp53 MO, and Δ113tp53 MO. TUNEL data of 48 hpf wild-type sibling and *banp^rw337* mutant retinas injected with STD MO and tp53 MO are shared with those shown in *Figure 3G*. tp53 MO-mediated knockdown completely rescues apoptosis in *banp^rw337* mutant retinas at 48 hpf. However, Δ113tp53 MO-mediated knockdown mildly inhibits apoptosis in *banp^rw337* mutant retinas. Neither tp53 MO nor Δ113tp53 MO effectively rescued mitotic cell accumulation and accumulation of γ-H2AX-positive cells (white arrowheads) in *banp^rw337* mutant retinas. Scale bars: 20 μm. (D–F) Histogram of the number of TUNEL-positive cells (D), pH3-positive cells (E), and γ-H2AX-positive cells (F) per retinal section in wild-type and *banp^rw337* mutants injected with STD-MO, tp53 MO, and Δ113tp53 MO. TUNEL data of 48 hpf wild-type sibling and *banp^rw337* mutant retinas injected with STD MO and tp53 MO are shared with those shown in *Figure 3H*. Two-way ANOVA with Tukey's multiple comparisons test, Mean ± SD. [n=in graph, p*<0.05, p**<0.01, p***<0.001, p****<0.0001, ns (not significant)]. (G) Possible role of Banp in tp53-mediated DNA damage response pathway. DNA replication stress activates ATR, which subsequently promotes DNA replication fork recovery. Failure of stalled fork recovery causes DSBs, which activate ATM-dependent DNA damage repair. ATR and ATM inhibit Mdm2-mediated tp53 degradation. Stabilized tp53 initially promotes transcription of the Δ113tp53 isoform, which activates cell-cycle arrest genes and promotes DNA damage repair. Chronic activation of tp53 promotes transcription of apoptotic genes to induce apoptosis. Banp may normally suppress DNA replication stress by promoting DNA replication fork recovery or DNA damage repair. Banp may suppress genotoxic stress-mediated activation of DNA replication stress and DSB formation.

The online version of this article includes the following source data and figure supplement(s) for figure 4:

**Source data 1.** Data for *Figure 4ADEF*.

**Figure supplement 1.** DNA damage is induced in retinal progenitor cells in *banp* knockdown.

**Figure supplement 1—source data 1.** Data for *Figure 4—figure supplement 1B, D, F, H, J*.

section) (*Figure 4C and F*). These data suggest that there is a tp53-independent mechanism that causes mitotic cell accumulation in *banp^rw337* mutants.

## Banp regulates chromatin segregation during mitosis by promoting transcription of *cenpt* and *ncapg* genes

A substantial fraction of accumulated mitotic cells in *banp^rw337* mutants is independent of tp53 (*Figure 4E*). RNA-seq analysis revealed that two chromosomal segregation regulators, *cenpt* and *ncapg*, are downregulated in *banp^rw337* mutants at 48 hpf (*Figure 3B*). Cenpt and Ncapg are essential for chromatin segregation (*Hung et al., 2017*; *Seipold et al., 2009*; *Zhang et al., 2018*). Using in situ hybridization, we confirmed that their mRNAs are expressed in retinal progenitor cells and that expression of both mRNAs is markedly decreased in *banp* mutants at 48 hpf (*Figure 5—figure supplement 1*). Quantitative RT-PCR also confirmed their downregulation in *banp^rw337* mutants and *banp* morphants (*Figure 5A and C*). Furthermore, ATAC-seq revealed that chromatin accessibility is decreased near the transcription start site (TSS) of both genes in *banp^rw337* mutants, leading to reduction of transcription activity (*Figure 5B and D*). Thus, Banp is required for transcriptional activation of *cenpt* and *ncapg* genes. Interestingly, two independent chip-sequencing studies using BANP antibody revealed a specific binding site within 100 bp of the TSS of *Cenpt* and *Ncapg* genes in mice and humans (*Grand et al., 2021*; *Mathai et al., 2016*), suggesting that the requirement for Banp in *cenpt* and *ncapg* transcription is conserved in zebrafish, mice, and humans.

Knockdown of Cenpt and Ncapg causes defects in chromosomal segregation during mitosis in zebrafish (*Hung et al., 2017*; *Seipold et al., 2009*). To examine whether *banp* mutants show similar chromosome segregation defects, we conducted time-lapse imaging of mitosis in *banp^rw337* mutant retinas for 1–2 hr from 48 to 54 hpf, using a transgenic line *Tg[h2afv:GFP; EF1α:mCherry-zGem]*, which visualizes cell-cycle phases (*Mochizuki et al., 2014*). In wild-type retinas injected with control banp 5misMO, chromatin starts to be condensed in prophase. Then, kinetochores and a mitotic spindle appear in prometaphase, chromosomes are centrally aligned and start to be pulled toward opposite poles by mitotic spindles in metaphase, sister chromatids move toward the opposite poles in anaphase, and chromosomes arrive at opposite poles in telophase, whereupon cytokinesis starts (*Figure 5E*, *Figure 5—video 1*). The duration from prophase to telophase was 22.9±5.8 minutes (*Figure 5F*). On the other hand, mitosis did not proceed smoothly in *banp* morphant retinas, especially from prophase to anaphase (*Figure 5E*, *Figure 5—video 2*). It took longer from chromosomal condensation in prophase to chromosomal alignment at the division plane in anaphase in *banp* morphant retinas (t=15 to t=46 in banp-MO, *Figure 5E*). Aligned chromosomes were often dispersed

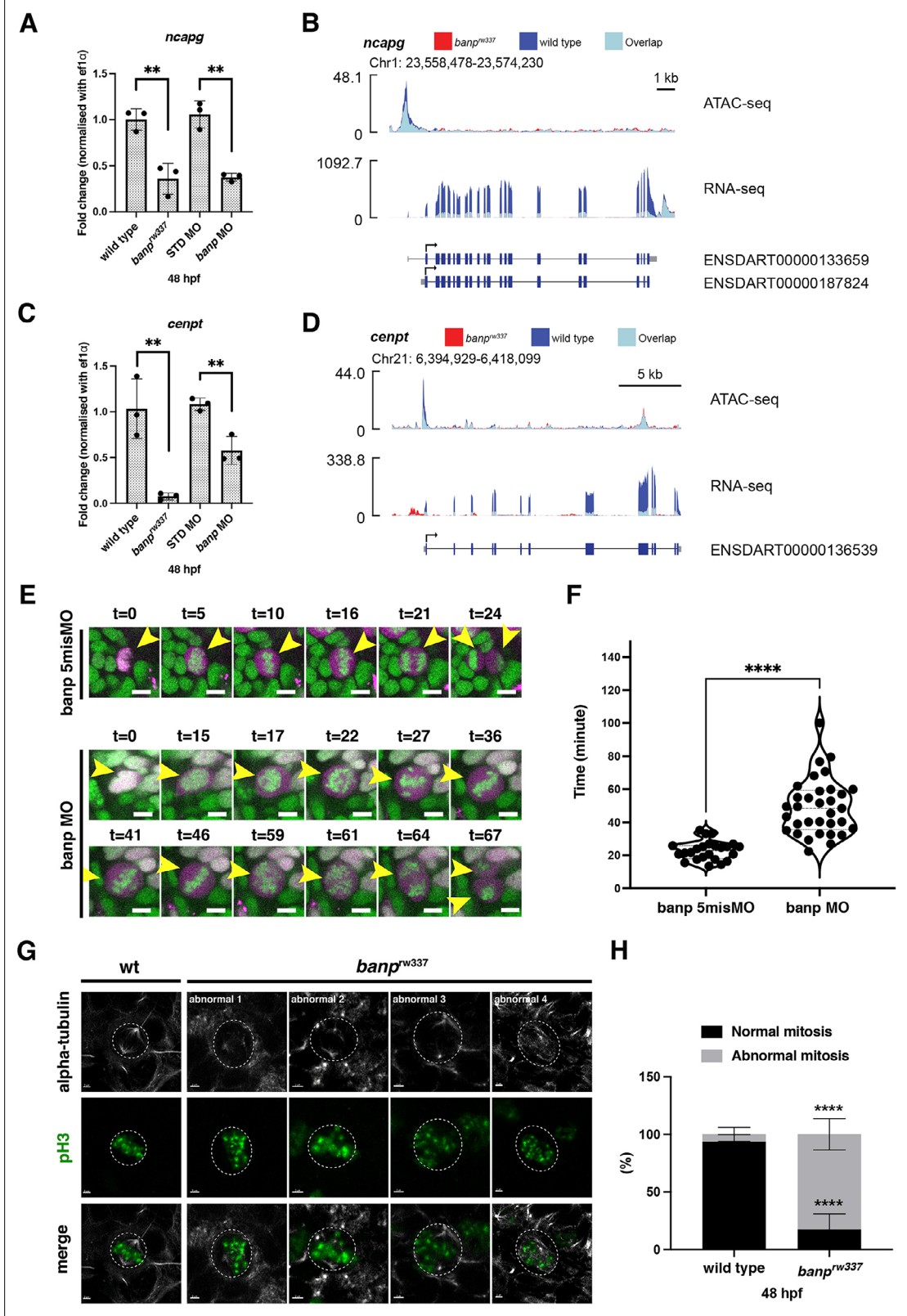

**Figure 5.** Banp promotes transcription of chromosome segregation regulators, *cenpt* and *ncapg*. (**A**) Quantitative RT-PCR of *ncapg* expression at 48 hpf. *ncapg* mRNA expression is down-regulated in *banp^rw337* mutants and *banp* morphants. Unpaired t-test (two-tailed), Mean ± SD. [n=3, p**<0.01]. (**B**) ATAC-seq and RNA-seq analysis of *ncapg* mRNA transcription. In *banp^rw337* mutants, chromatin access is reduced at the TSS of *ncapg* (ATAC-seq) and transcript expression is markedly decreased (RNA seq). (**C**) Quantitative RT-PCR of *cenpt* expression at 48 hpf. *cenpt* mRNA expression is down-

*Figure 5 continued on next page*

*Figure 5 continued*

regulated in *banp^rw337* mutants and *banp* morphants. Unpaired t-test (two-tailed), Mean ± SD. [n=3, p**<0.01]. (**D**) ATAC-seq and RNA-seq analysis of *cenpt* mRNA transcription. In *banp^rw337* mutants, chromatin access is reduced at the TSS of *cenpt* (ATAC-seq) and transcript expression is markedly decreased (RNA seq). (**E**) Time-lapse images of mitosis of retinal progenitor cells injected with banp 5misMO (upper) and banp MO (lower) at 2 dpf using *Tg [h2afv: GFP; EF1α: mCherry-zGem]* transgenic zebrafish. In retinas injected with banp 5misMO, mitosis proceeds sequentially from prophase (t=5–10 min), metaphase (t=16 min), anaphase (t=21 min), and telophase (t=24 min). On the other hand, in *banp* morphant retinas, the duration of mitosis from prophase to telophase was 67 min, significantly longer than that of the control MO (24 min). Furthermore, a prophase-like chromosome arrangement appeared abnormally (t=59) after several metaphase-like alignments (t=36 and 46), followed by anaphase (t=64) and telophase. Scale bar: 5 μm. (**F**) Violin plot of the duration of mitosis from prophase to telophase in retinas injected with banp 5misMO and banp MO. Mitosis in *banp* morphants is significantly longer than that of banp 5misMO injected embryos. Mann Whitney test, Mean ± SD. [n=3, p****<0.0001] (**G**) Confocal images of metaphase cells in zebrafish retinas at 48 hpf. The mitotic spindle is labeled with anti-α-tubulin antibody, whereas metaphase chromosomes are labeled with anti-pH3 antibody. The mitotic spindle is correctly attached to metaphase chromosomes in wild-type retinal cells (left panel). On the other hand, mitotic cells in *banp^rw337* mutants exhibit inefficient mitotic spindle attachment and poorly orientated metaphase chromosomes (right panels). Dotted circles indicate the outline of metaphase cells. Scale bar: 2 μm. (**H**) Percentage of cells with normal and abnormal spindle attachment in wild-type and *banp^rw337* mutant embryos. Two-way ANOVA with Šídák's multiple comparisons test, Mean ± SD. [n=3, p****<0.0001].

The online version of this article includes the following video, source data, and figure supplement(s) for figure 5:

**Source data 1.** Data for *Figure 5ACFH*.

**Figure supplement 1.** Spatio-temporal pattern of zebrafish *cenpt* and *ncapg* mRNA expression.

**Figure 5—video 1.** Time-lapse movie of mitosis of wild-type retinal progenitor cells.

https://elifesciences.org/articles/74611/figures#fig5video1

**Figure 5—video 2.** Time-lapse movie of mitosis of *banp* morphant retinal progenitor cells.

https://elifesciences.org/articles/74611/figures#fig5video2

---

during chromosomal segregation by the mitotic spindle (t=46 to t=64 in banp-MO, *Figure 5E*), although all 28 cell divisions eventually completed cytokinesis in *banp* morphants. The duration from prophase to telophase was 49.2±17.0 min and significantly longer in *banp* morphants (*Figure 5F*). Thus, chromosomal segregation is abnormal during mitosis in the absence of Banp. Furthermore, we labeled 48-hpf retinas with anti-α-tubulin and anti-pH3 antibodies to visualize the mitotic spindle and prometaphase chromosomes, respectively. In *banp^rw337* mutants, mitotic spindles are irregularly attached to less condensed and misaligned chromosomes (*Figure 5G*). The percentage of cells with abnormal mitosis was significantly higher in *banp^rw337* mutant retinas than those of wild-type siblings (*Figure 5H*). These data suggest that the interaction between chromosomes and mitotic spindles is compromised in the absence of Banp, probably due to kinetochore attachment defects caused by lack of Cenpt and Ncapg. Thus, Banp regulates chromosomal segregation during mitosis.

## Banp regulates multiple gene targets as a transcription activator through the Banp motif during embryonic development

Recently, it was reported that BANP binds to the nucleotide sequence TCTCGCGAGA, called the 'Banp motif' and regulates transcription of essential metabolic genes in humans and mice (*Grand et al., 2021*). To identify direct transcriptional targets of Banp in zebrafish, we examined whether genes whose mRNA level or chromatin accessibility is decreased in *banp* mutants carry Banp motif in their promoters. So, we first searched for enriched motifs in *banp^rw337* mutants using a motif discovery software called Hypergeometric Optimization of Motif EnRichment (HOMER) with ATAC-seq data. Consistent with a previous report on human and mouse BANP (*Grand et al., 2021*), our analysis revealed that TCTCGCGAGA, is the most enriched motif in the closed chromatin region of *banp^rw337* mutants (*Figure 6A* and *Figure 6—figure supplement 1A*). Indeed, among 81 down-regulated genes in *banp^rw337* mutants (*Figure 3A*), we identified the Banp motif near TSS of 31 genes (*Figure 6B and C* and *Supplementary file 1*). However, we could not find the Banp motif among 258 upregulated genes (*Figure 6—figure supplement 1A*). These data suggest that Banp is a transcription activator of multiple gene targets containing Banp motifs. The position of the Banp motif is consistent with the previous report, that it is not located at a constant distance from TSSs, but rather between 20 and 70 nucleotides upstream on both strands. We also observed multiple Banp motifs in some genes (*Supplementary file 1*). It is likely that the frequency of Banp motifs upstream of TSSs may correlate with a transcription rate of target genes (*Mahpour et al., 2018*).

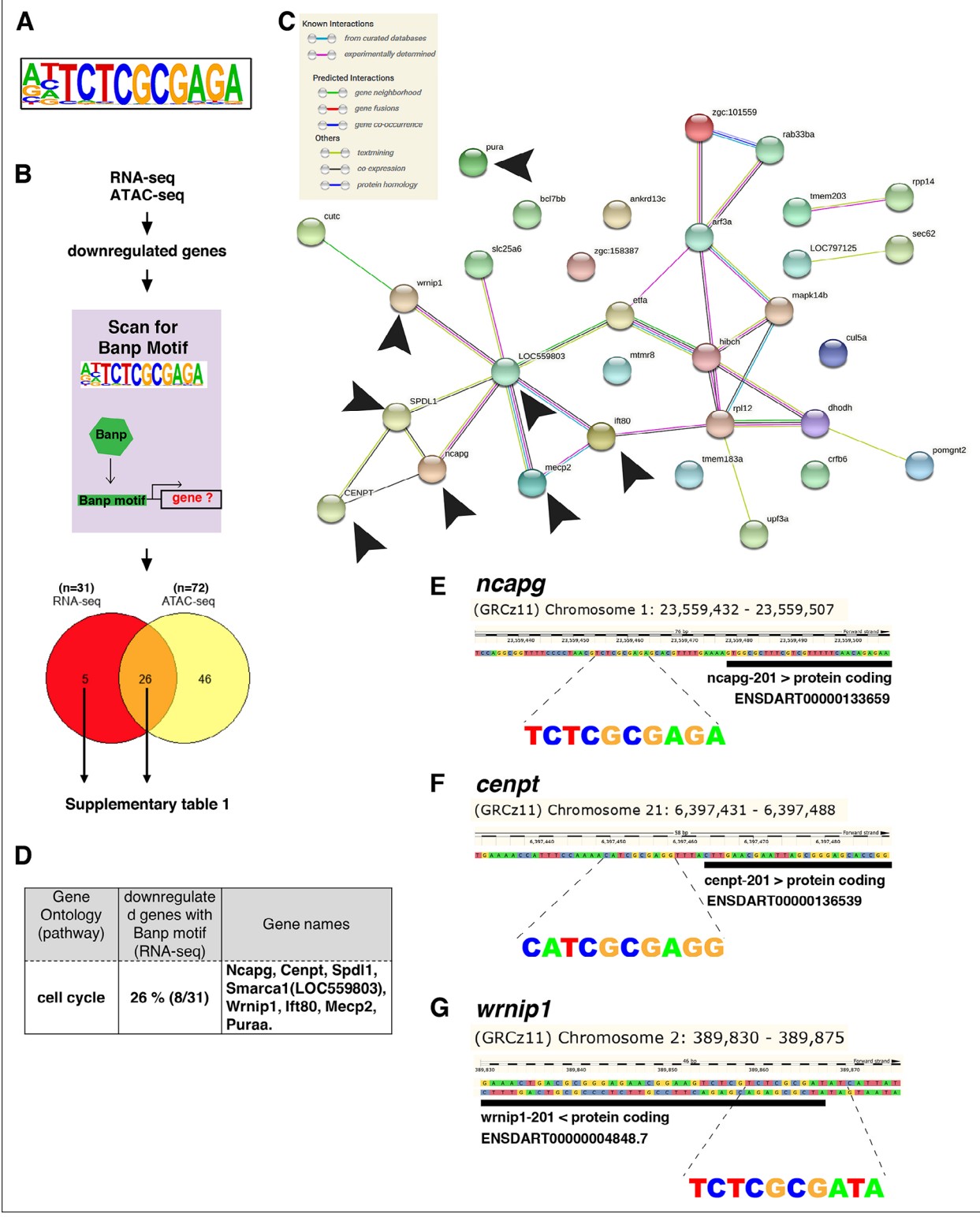

**Figure 6.** Banp promotes expression of genes containing Banp motifs. (**A**) The motif enriched in genomic regions in which chromatin accessibility is reduced in *banp^rw337^* mutants. This sequence shows a high similarity to the Banp motif. (**B**) A schematic diagram representing the flow of RNA-sequencing combined with ATAC-sequencing to find candidates for zebrafish Banp direct-target genes. HOMER was used to scan for Banp motifs in downregulated candidate genes, from RNA- and ATAC-sequencing in *banp^rw337^* mutants. RNA sequencing yielded 31 candidates, whereas ATAC-sequencing yielded 72, with 26 target genes consistent in both. These 26 candidate genes are the most likely to be target genes of Banp. (**C**) STRING interactome analysis showing the interaction of Banp motif-containing genes downregulated in RNA-sequencing. Black arrowheads specify genes

*Figure 6 continued on next page*

*Figure 6 continued*

related to cell-cycle regulation. (**D**) Banp target candidate genes related to cell-cycle regulation. Gene ontology (GO) analysis was applied to 31 Banp motif-containing genes downregulated in RNA-sequencing. Twenty-six percent of these genes are engaged in the cell-cycle regulation. Others participate in a variety of non-categorized pathways. (**E**) Banp motif upstream to the 5'-UTR of *ncapg*. The black box represents the 5'-UTR. (**F**) Banp motif upstream to the 5'-UTR of *cenpt*. The black box represents the 5'-UTR. (**G**) Banp motif upstream to the 5'-UTR of *wrnip1*. The black box represents the 5'-UTR.

The online version of this article includes the following source data and figure supplement(s) for figure 6:

**Figure supplement 1.** Motifs enriched in closed and open chromatin areas in *banp^{rw337}* mutants.

**Figure supplement 2.** Banp promotes transcription of a DNA replication regulator *wrnip1*.

**Figure supplement 2—source data 1.** Data for *Figure 6—figure supplement 2C*.

Next, we examined profiles of these direct-target genes. Among the transcription targets of Banp that are downregulated in *banp^{rw337}* mutants, 26 percent were involved in regulating cell cycle (*Figure 6D*), including *ncapg* (*Figure 6E* and *Figure 6—figure supplement 1B*), *cenpt* (*Figure 6F* and *Figure 6—figure supplement 1C*), and *wrnip1* (*Figure 6G* and *Figure 6—figure supplement 2A*). These data again confirmed that Banp profoundly regulates chromosome segregation during mitosis by directly activating transcription of *ncapg* and *cenpt* genes. In addition, *wrnip1* is interesting, because it protects stalled replication forks from MRE11-mediated degradation and promotes fork restart after replication stress (*Leuzzi et al., 2016a*; *Leuzzi et al., 2016b*). We confirmed that *wrnip1* mRNA is normally expressed in retinal progenitor cells in zebrafish and that its expression is reduced in *banp^{rw337}* mutants at 48 hpf (*Figure 6—figure supplement 2B*). Quantitative RT-PCR analysis revealed a significant reduction of *wrnip1* mRNA expression in both *banp* mutants and morphants (*Figure 6—figure supplement 2C*). Consistently, RNA-seq data showed downregulation of *wrnip1* (*Figure 6—figure supplement 2D*), although ATAC-seq data showed comparable expression in wild-type and mutants. These data suggest that Banp is required to maintain *wrnip1* mRNA expression in retinal progenitor cells in zebrafish. Loss of *wrnip1* is expected to cause genomic instability due to excessive fork degradation (*Leuzzi et al., 2016b*). So, the reduction of *wrnip1* mRNA expression may explain the significant accumulation of γ-H2AX+ retinal cells in *banp^{rw337}* mutants (*Figure 4—figure supplement 1C,D*). Taken together, these data suggest that Banp regulates transcription of multiple target genes, including these cell-cycle regulators.

In summary, Banp serves essential functions in cell-cycle regulation in the zebrafish developing retina (*Figure 7A*). During S phase, Banp suppresses DNA replication stress by activating transcription of a DNA replication stress regulator, *wrnip1*, and promotes DNA damage repair in concert with tp53-mediated DNA damage response. In M phase, Banp promotes mitosis by activating transcription of two chromosome segregation regulators, *cenpt* and *ncapg*, most likely via their Banp motifs. Thus, Banp regulates progression in S phase and M phase through transcriptional control of distinct sets of cell-cycle regulators. In *banp* mutants, stalled DNA replication and subsequent DSBs are accumulated in S phase, resulting in increase in apoptosis. In M phase, *banp* mutants show chromosome segregation defects, which prolongs mitosis but eventually complete it. So, DNA damage mostly induced in S phase is likely to be inherited through mitosis by daughter cells, which reenter the cell-cycle or become post-mitotic neurons. Neurons with a high level of DNA damage may undergo apoptosis. Thus, S phase defects and M phase defects influence each other along cell-cycle progression of retinal progenitor cells in the absence of Banp (*Figure 7B*).

## Discussion

BANP was initially identified as a nuclear MAR-binding protein and reported to function as a tumor suppressor (*Malonia et al., 2011*). Very recently, it was reported that BANP binds to the Banp motif enriched near the TSS of CpG island promoters and regulates transcription of metabolic genes (*Grand et al., 2021*). However, physiological functions of Banp remained to be discovered. In this study, we identified zebrafish *banp* mutants, in which retinal progenitor cells show mitotic defects and apoptosis during embryonic development. RNA-seq analysis revealed that expression of tp53-dependent DNA damage response regulators is upregulated in *banp* mutants. In non-stress conditions, tp53 is degraded through ubiquitination by an E3-ubiqutin ligase, Mdm2, and subsequent actions of proteasomes, keeping its protein level very low (*Oliner et al., 1993*). On the other hand, under stress, such

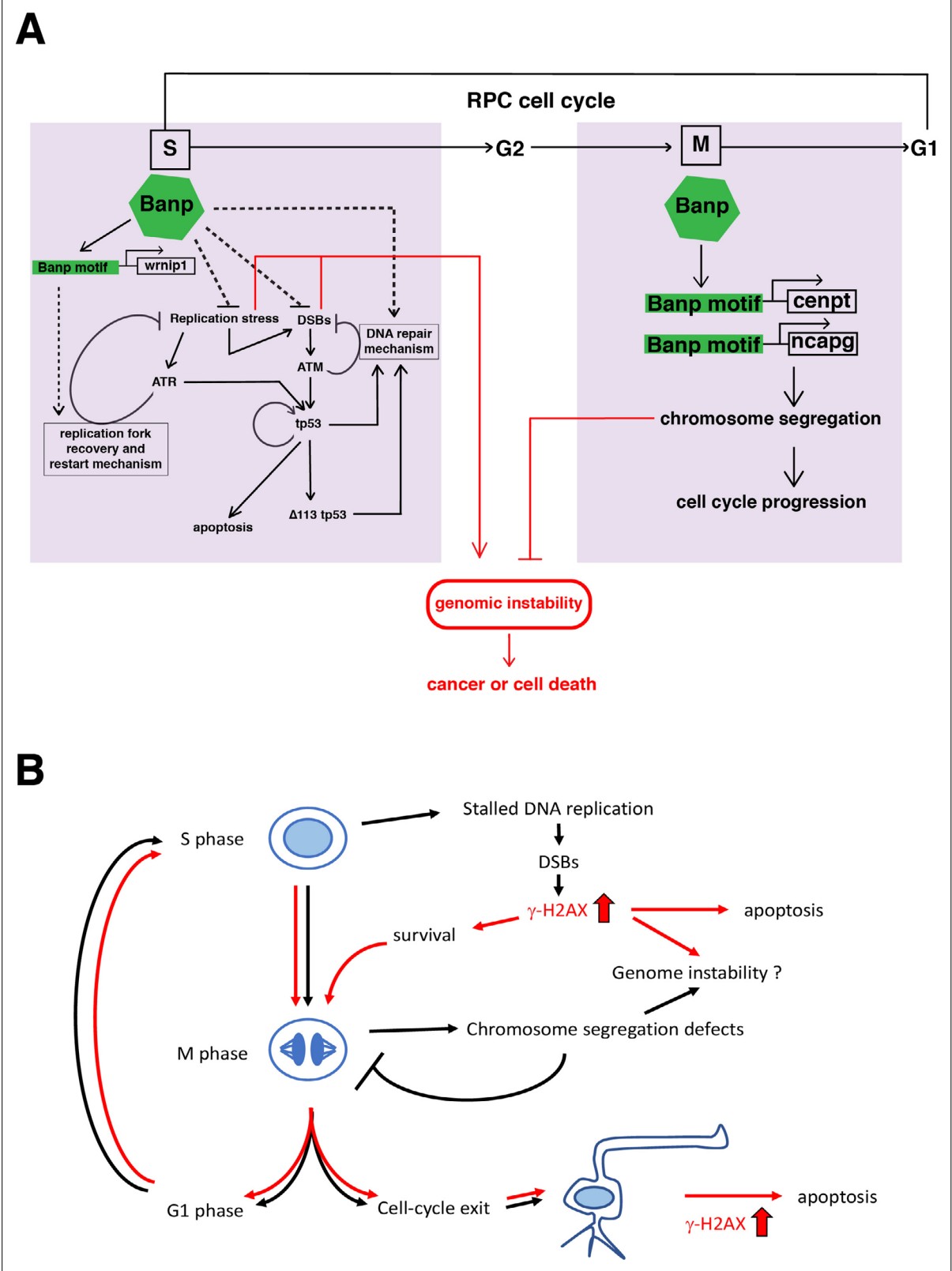

**Figure 7.** A possible model of Banp functions in stalled fork recovery, DNA damage repair, and chromosome segregation during mitosis. (**A**) Our study reveals two major roles of Banp in cell-cycle regulation of retinal progenitor cells. In general, DNA replication stress, DSBs, and chromosome segregation defects induce genomic instability, which causes cancer or cell death. In S phase, Banp may suppress DNA replication stress by promoting transcription of the *wrnip1*, which protects stalled replication forks and promotes replication restart after replication stress. Failure of recovery from

*Figure 7 continued on next page*

*Figure 7 continued*

DNA replication stress induces DSBs, which activate ATM/Chk2/tp53-mediated DNA damage response. Banp may promote DNA damage repair in concert with tp53. ATR also activates tp53 through activation of Chk1 or crosstalk with ATM. Banp may suppress genotoxic stress-mediated DNA replication stress and DSBs. In M phase, Banp is required for chromosome segregation by promoting transcription of two mitotic regulators, *cenpt* and *ncapg*. Since a Banp motif is found near TSSs of the *cenpt*, *ncapg* and *wrnip1* genes, it is likely that these genes are direct targets of Banp. Dotted arrows indicate proposed regulatory pathway, which will be necessary to investigate in the future. (**B**) *banp^rw337* mutant phenotypes along the cell-cycle progression and neurogenesis. In *banp* mutants, stalled DNA replication and subsequent DSBs are accumulated, leading to increase in γ-H2AX+ cells. These DNA damaged cells undergo apoptosis. In M phase, mitosis is markedly prolonged due to chromosome segregation defects; however, almost all mitoses are eventually completed, so it is less likely that M phase defects directly link to apoptosis. It is possible that a fraction of γ-H2AX+ cells survive and enter M phase, so DNA damage is inherited by daughter cells, which reenter the cell-cycle or differentiate into postmitotic neurons. Neuron with a high level of γ-H2AX+ signals may undergo apoptosis. Thus, S phase defects and M phase defects influence each other through cell-cycle progression in *banp* mutants. Red lines indicate path of γ-H2AX+ cells.

as DNA replication stalling or DSBs, ATR, and ATM are activated and phosphorylate the N-terminal serine of tp53 (*Banin et al., 1998*; *Canman et al., 1998*; *Siliciano et al., 1997*; *Tibbetts et al., 1999*). This phosphorylation reduces the binding affinity of Mdm2 for tp53, resulting in stabilization of tp53 protein (*Shieh et al., 1997*; *Yadahalli et al., 2019*). Indeed, the tp53 protein level is increased, and tp53 knockdown effectively suppresses retinal apoptosis in *banp* mutants. Furthermore, the number of γ-H2AX+ cells is significantly increased in *banp* mutants, suggesting elevation of replicative DNA damage. These data suggest that DNA replication stress is elevated in *banp* mutants, resulting in activation of tp53-dependent DNA damage response, which subsequently induces apoptosis of retinal progenitor cells.

It is generally accepted that tp53 has dual, but opposing roles: cell survival versus cell death in response to DNA replication stress and DNA damage (*Chen, 2016*; *Lees et al., 2021*). In the initial phase of the DNA damage response, tp53 binds to its alternative promoter located in intron 4 and activates transcription of an N-terminal truncated isoform, Δ113-tp53, which preferentially activates transcription of target genes related to cell-cycle arrest to promote DNA repair and cell survival (*Chen et al., 2009*). However, if DNA damage fails to be repaired, but accumulates chronically, FL tp53, in turn, activates transcription of apoptotic genes to eliminate DNA-damaged cells. Previous in vitro studies of human *BANP* suggest that BANP interacts with tp53 to promote transcription of cell-cycle arrest genes and to suppress transcription of apoptotic genes, leading to promotion of its cell survival role (*Sinha et al., 2010*). Indeed, our quantitative RT-PCR showed that transcription of Δ113-tp53 is significantly elevated in *banp* mutants, whereas the mRNA level of FL tp53 is very low. RNA-seq and ATAC-seq also confirmed a higher level of transcription and chromatin accessibility of Δ113tp53 in *banp* mutants. These data suggest that Δ113tp53-mediated transcription is selectively activated in *banp* mutants. Thus, it is important to clarify the role of tp53, including Δ113tp53 in mitotic cell accumulation in *banp* mutants, because pH3+ cells are markedly accumulated in *banp* mutant retinas prior to apoptosis. However, tp53-MO mildly inhibited accumulation of pH3+ cells in *banp* mutants. There is no significant difference in pH3+ cell accumulation between tp53-MO and Δ113tp53-MO, or between wild-type siblings and Δ113tp53-MO. So, these data suggest that the contribution of tp53 to mitotic cell accumulation in *banp* mutants is limited, and that a tp53-independent pathway is responsible for mitotic phenotypes. Our live imaging of mitosis in *banp* morphant retinas reveals that mitosis is significantly longer due to chromosome segregation defects; however, importantly, almost all mitoses are eventually completed in *banp* mutants, suggesting that accumulation of pH3+ cells in *banp* mutants is transient and depends mostly on prolonged mitosis, which is caused by chromosome segregation defects. At present, we have not observed S-phase arrest in *banp* mutants, because the fraction of BrdU+ cells in the total retinal region is not increased in *banp* mutants, although it is still possible that *banp* mutant retinal cells are partially arrested in G1 phase, because mRNA expression of a cdk inhibitor, *cdkn1a* (*p21*), is higher in *banp* mutants than in wild-type siblings. Taken together, we conclude that tp53 mainly mediates retinal apoptosis, and that its role in mitotic defects is limited in *banp* mutants.

Labeling with anti-γ-H2AX antibody revealed that replicative DNA damage is increased in retinal progenitor cells in *banp* mutants, suggesting that DNA lesions that cause stalled DNA replication are abnormally induced, or stalled DNA replication and DSBs fail to be repaired in the absence of Banp. Importantly, the number of γ-H2AX+ cells in *banp* mutant retinas was not increased by either tp53-MO and Δ113tp53-MO, suggesting that tp53 does not repair replicative DNA damage caused

by loss of Banp functions. There are two possible functions of Banp in DNA damage regulation. Banp may maintain DNA replication integrity. BANP functions as a transcription factor to regulate expression of essential metabolic genes in pluripotent stem cells and differentiating neurons (*Grand et al., 2021*). Banp may promote transcription of DNA replication regulators. Our RNA-seq data identified 81 downregulated genes in *banp* mutants, compared to wild-type siblings. Among them, 31 genes have a Banp motif near their TSSs and contain eight cell-cycle regulatory genes. Among these eight cell-cycle regulators, interestingly, we identified *wrnip1*. WRNIP1 protects stalled replication forks, especially fork reversal, from MRE11- and SLX4-mediated degradation and promotes fork restart by cooperating with BRCA2 to stabilize RAD51 on single-stranded DNA of stalled forks (*Leuzzi et al., 2016a*; *Leuzzi et al., 2016b*; *Porebski et al., 2019*). In addition, *wrnip1* regulates another fork protection mechanism called damage bypass, consisting of translesion synthesis (TLS) (*Yoshimura et al., 2019*) and template switching (TS) (*Jiménez-Martín et al., 2020*). WRNIP1 also functions in interstrand crosslink repair (ICR) by promoting the recruitment of FANCD2/FANCI complex at the damaged site (*Socha et al., 2020*). Indeed, *wrnip1* mRNA expression is markedly reduced in *banp* mutants, suggesting that Banp is required for maintenance of *wrnip1* transcription. In this case, loss of Wrnip1 may cause defects in stalled fork stabilization and ICR repair, leading to genomic instability (*Kondratick et al., 2021*; *Nickoloff et al., 2021*; *Scully et al., 2021*). It will be interesting to examine whether knockdown of zebrafish *wrnip1* causes similar defects in DNA replication stress and apoptosis to *banp* mutants. Another possibility is that Banp regulates DNA repair independent of its role as a transcription factor. A previous study using human in vitro culture cells revealed that BANP promotes HDAC6-mediated deacetylation of a DNA end-joining protein, Ku70, which is required for non-homologous end-joining repair (*Chaudhary et al., 2014*). Since Ku70 is recruited to DSB sites after γ-H2AX accumulation, Ku70 is an interesting candidate. Since knockdown of Ku70 along with radiation-induced DNA damage markedly induces apoptosis in zebrafish embryos (*Bladen et al., 2007*), it will be interesting to investigate whether this scenario occurs in zebrafish.

In addition to the role of Banp in DNA replication stress and DNA repair mechanisms, mRNA expression of two important regulators of chromosome segregation, *ncapg* and *cenpt*, is significantly downregulated in *banp* mutants. ATAC-seq and RNA-seq confirmed that chromatin accessibility is decreased near the TSSs of these genes in *banp* mutants, suppressing transcription of these mRNAs. Our de novo motif enrichment study identified the Banp motif sequence, TCTCGCGAGA, as the most enriched motif of closed chromatin regions of downregulated genes. Thus, it is highly likely that Banp directly binds to the Banp motifs of both *ncapg* and *cenpt*, opens their chromatin, and promotes their transcription. Consistently, as reported in zebrafish *ncapg* and *cenpt* knockdown embryos (*Hung et al., 2017*; *Seipold et al., 2009*), chromosome segregation is compromised in mitosis of retinal progenitor cells in *banp* mutants, suggesting that Banp is an essential regulator of mitosis. Since a Banp motif is found in promoter regions of *Ncapg* and *Cenpt* in mice and humans, and Chip sequencing with BANP antibody validated TSS peaks of both *Ncapg* and *Cenpt* genes in mice and humans (*Grand et al., 2021*; *Mathai et al., 2016*), it will be interesting to confirm whether BANP regulates chromosome segregation during mitosis in mammals.

Lastly, we identified 26 candidate Banp direct-target genes, including cell-cycle regulators such as *ncapg, cenpt* and *wrnip1*, which carry Banp motifs near their TSSs, and show markedly decreased chromatin accessibility and transcription in the absence of Banp. This gene list will provide a useful starting point to investigate the Banp-mediated transcriptional network. It was reported that binding of BANP to Banp motifs is dependent on DNA methylation of Banp motifs (*Grand et al., 2021*), suggesting that an epigenetic mechanism regulates Banp-mediated transcription. Thus, zebrafish *banp* mutants provide an excellent new model in which to study how epigenetic factors regulate cellular processes for embryonic development via Banp motifs of CpG island promoter genes. In addition, BANP has been proposed as a tumor suppressor (*Malonia et al., 2011*). It is generally accepted that DNA methylation of essential genes is altered in cancer cells (*Baylin et al., 2001*), which is expected to modify the pattern of gene transcription through Banp motifs (*Grand et al., 2021*). It will be interesting to investigate how the Banp-mediated transcription profile is changed in cancer models of zebrafish. Future research on zebrafish *banp* mutants will provide more in-depth understanding of cancer growth and metastasis, as well as new insights into development of therapeutics for human tumors.

# Materials and methods

## Zebrafish strains and maintenance

Zebrafish (*Danio rerio*) were maintained using standard procedures (*Westerfield, 1993*). RIKEN Wako (RW) was used as a wild-type strain for mutagenesis in which *banp*[rw337] was generated. Mapping of the *rw337* mutation and analysis of retinal phenotypes of *banp*[rw337] mutants were carried out in the genetic background of WIK and Okinawa wild-type (oki), respectively. Two *banp* mutant alleles, *banp*[rw337] and *banp*[sa12976] were used. *banp*[sa12976] was obtained from the Zebrafish International Resource Center (ZIRC, Eugene, OR, USA). Zebrafish transgenic lines, *Tg[h2afv:GFP]*[kca6/kca66] (*Pauls et al., 2001*), *Tg[EF1α:mCherry-zGem]*[oki011] (*Mochizuki et al., 2014*), *Tg[ath5:EGFP]*[w021] (*Masai et al., 2005*) and *Tg[EF1α:mCherry-CAAX]*[oki049] (*Mochizuki et al., 2017*) were used to visualize chromatin, S/G2/M phases, RGCs/amacrine cells/photoreceptors, and plasma membranes, respectively.

## Zebrafish mutagenesis, mapping and cloning of the *banp* mutant gene

Mutagenesis, mapping, and cloning were performed as previously described (*Masai et al., 2003*). The polymorphic markers, 070702B and zC93F2D, shown below were designed and used for restricting the genomic region covering the *banp*[rw337] mutation.

070702B forward: 5'-ACTTCTTATCAGGGCTGTGC-3'; 070702B reverse: 5'-TCAGTCAAGAGCA GTGAGAG-3'; zC93F2D forward: 5'-TGGGATCTCTTTAAGTGAGTGAG-3'; zC93F2D reverse: 5'-T CCAACTATGTGGGTCAAACC-3'.

## Genotyping of *banp*[rw337] mutant embryos

*banp*[rw337] mutant phenotypes are difficult to distinguish until 2 dpf. Identification of homozygous mutant embryos at earlier time points was performed using PCR amplification of genomic regions using specific primers that enabled restriction digestion of only mutated sequence using MboII. Primers used, forward: 5'-CGATGTTGATATCCATCAGTCAGGCGATC-3'; reverse primer: 5'-GGTGC TGGTGTATAAATCACATGACCTATGGTCCTCTT-3'.

## Acridine orange staining

53-hpf embryos were incubated at 28 °C, in 5 mg/mL Acridine Orange (AO) in E3 embryonic medium (5 mM NaCl, 0.17 mM KCl, 0.33 mM $CaCl_2$, 0.33 mM $MgSO_4$) for 30 min, protected from light. Embryos were rinsed in E3 medium and used for detection of AO+ signals. Retinas of live embryos were scanned from the lateral view with a confocal laser scanning microscope (LSM) (LSM710, Zeiss). After scanning of wild-type embryos, *banp* mutants, and *banp* mutants injected with mRNA encoding banp(wt)-EGFP and EGFP-banp(rw337), these scanned embryos were used for DNA extraction and genotyping. Five samples were selected from each group and used for statistical analysis.

## In situ hybridization

Whole-mount in situ hybridization of zebrafish embryos was performed as previously described (*Xu et al., 1994*). To prepare RNA probes, full-length cDNAs of the zebrafish *banp* gene were amplified with Polymerase Chain Reaction (PCR) and sub-cloned into pBluescript II SK(+) vectors (Stratagene/ Agilent Technologies). cDNA fragments were amplified from mRNA with PCR using the primers for *cenpt*, *ncapg* and *wrnip1*, and subcloned into pCRII TOPO vector. All the primer sequences of *banp*, *cenpt*, *ncapg*, and *wrnip1* for PCR amplification are provided in Key Resources Table. Plasmids were digested with an appropriate restriction enzyme to make DNA templates, which were used to make antisense or sense RNA probes. Digoxigenin-labeled RNA probes were generated by in vitro transcription using a DIG RNA Labeling Kit (Roche) and purified using Micro Bio-Spin Chromatography Columns (BioRad). In situ hybridization signals in whole-mount embryos were detected using anti-DIG alkaline phosphatase (AP)-conjugated antibody (Roche) with an AP substrate, BM-Purple (Roche) and imaged using an SteREO Discovery.V12 (ZEISS) microscope.

## Rescue experiment with *banp* mutants overexpressing wild-type Banp

The full-length coding region of the *banp* gene was amplified with PCR. To overexpress wild-type Banp protein, a DNA construct encoding zebrafish wild-type Banp with an EGFP tag at the N-terminus or C-terminus, EGFP-banp(wt) or banp(wt)-EGFP, was prepared. A DNA construct encoding an

N-terminal EGFP-tagged *rw337* mutant form of Banp, EGFP-Banp(rw337), was prepared as a negative control. These DNA constructs were subcloned into the pCS2 expression vector. Capped mRNA was generated in a standard in vitro transcription reaction using an SP6 polymerase (mMESSAGE mMACHINE SP6 Transcription Kit, Invitrogen). We performed mRNA injections in one-cell-stage embryos obtained from pair-wise crosses of *banp*[rw337] heterozygous mutant male and female fish. At 2 dpf, embryos were used to determine their genotype with PCR and to identify homozygous mutants. Plastic sections were obtained to evaluate retinal phenotypes of *banp*[rw337] mutant embryos injected with mRNA encoding banp(wt)-EGFP, or EGFP-banp(rw337). Furthermore, we conducted AO labeling of wild-type embryos, *banp*[rw337] mutants, *banp*[rw337] mutants expressing banp(wt)-EGFP, and *banp*[rw337] mutants expressing EGFP-banp(rw337). The fractional AO+ area of the total area of retinal confocal sections was measured using the LSM software, ZEN (Zeiss), to evaluate the rescue level. We also evaluated protein stability and nuclear localization of wild-type and *rw337* mutant form of Banp by live confocal imaging of wild-type retinas expressing banp(wt)-EGFP, EGFP-banp(wt), or EGFP-banp(rw337) at 24 hpf.

## Morpholino injection

ATG morpholino antisense oligos for *banp* (banp MO) and 5 mismatch *banp* MO (banp 5misMO) were designed as 5′-CCACTAAATCTTGCTCTGACATCAT-3′ and 5′-CCtCaAAATgTTcCTCTcACATCAT -3′, respectively. Morpholino antisense oligos for tp53 (tp53 MO) (5′-GCGCCATTGCTTTGCAAGAAT TG-3′) (*Langheinrich et al., 2002*), standard MO (STD MO) (5′-CCTCTTACCTCAGTTACAATTTATA-3′) were used. To knock down Δ113tp53, morpholino antisense oligos for Δ113tp53 (Δ113tp53 MO; also referred to as MO6-tp53 in the zebrafish database ZFIN) (5′-GCAAGTTTTTGCCAGCTGACAGAAG-3′ ) was used (*Chen et al., 2009*). All MOs were injected into one-cell-stage eggs at 1 mM.

## Histology

Plastic sectioning, immunostaining of cryosections, paraffin sectioning, and BrdU incorporation were performed as described previously (*Imai et al., 2010*), except that we injected BrdU solution into the yolk of embryos at 48 hpf instead of soaking embryos in 10 mM BrdU solution. Anti-pH3 antibody (Cell signalling technology, 6G3, 9706; Sigma-Aldrich, 06–570), anti-Pax6 antibody (Covance; PRB-278P), zpr1 antibody (ZIRC, Eugene, Oregon), anti-Prox1 antibody (GeneTex, GTX128354), anti-glutamine synthetase (GS) (Millipore, MAB302), anti-BrdU antibody (abcam, Ab6326), anti-PCNA (Sigma-Aldrich, P8825) were used at 1:500, 1:500, 1:100, 1:500, 1:100, 1:200, 1:200 dilutions, respectively. Nuclear staining was performed using SYTOX Green (Molecular Probes). Filamentous actin (F-actin) was stained using 0.1 mM rhodamine-conjugated phalloidin (Molecular Probes). TUNEL was performed using the In Situ Cell Death Detection Kit (Roche).

Whole-mount immunostaining with anti-α-tubulin antibody (Sigma-Aldrich; #T5168) to label the mitotic spindle was performed as follows. Homozygous mutant embryos were selected by AO staining, and fixed in 2% trichloroacetic acid (TCA) at room temperature (RT) for 3 hr. Then embryos were washed three times in 1 X PBST (PBS, 0.1% Triton X-100), incubated in 0.2% trypsin for 4 min at 4 °C to increase antibody permeabilization through the skin, and rinsed three times with PBS. After embryos were re-fixed with 4% PFA for 5 min at 4 °C, they were incubated in blocking solution (10% Goat serum in PBST) for 1 hr at room temperature and transferred to 1% blocking solution with diluted anti-α-tubulin antibody at a 1:1,000 dilution for 2 days at 4 °C. Embryos were washed with PBST four times for 15 min and then incubated with fluorescent conjugated anti-mouse antibody in 1% blocking solution in the dark for 2 days at 4 °C. Next, all samples were washed in PBST three times for 15 min followed by 70% glycerol storage at 4 °C until imaging. Samples were imaged with a confocal microscope (LSM 880, Zeiss) followed by image analysis using IMARIS software (ver.9.1.2 Bitplane). Whole-mount immunostaining for anti-pH3 and anti-γ-H2AX (GeneTex, GTX127342) antibodies were performed as described previously (*Sidi et al., 2008*), and for anti-active caspase 3 (BD Pharmingen, Clone C92-605) and anti-HuC/D (Thermo Fisher, A-21271) antibodies as described (*Sorrells et al., 2013*), except using goat serum as a blocking reagents. Images were scanned using an LSM710 confocal laser scanning microscope (Zeiss). The number of fluorescent-positive cells and percentage of fluorescence-positive area relative to the region of interest was calculated using original scanning images with IMARIS (Bitplane) and Zen (Zeiss) or Image-J (NIH) software, respectively.

## TUNEL of wild-type sibling and *banp*^rw337^ mutant retinas injected with STD MO, FL tp53 MO, and Δ113tp53 MO

We performed three MO experiments (STD-MO, FL tp53-MO, Δ113tp53-MO injection to wild-type sibling and *banp*^rw337^ mutant embryos) at 48 hpf and two MO experiments (STD-MO, FL tp53-MO injection to wild-type sibling and *banp*^rw337^ mutant embryos) at 72 hpf. After genotyping of all the injected embryos, wild-type and *banp*^rw337^ mutant embryos were selected for TUNEL. First, STD-MO and FL tp53-MO data at 48 and 72 hpf were used for statistatical analysis to compare inhibition levels of retinal apoptosis between STD-MO and FL tp53-MO (*Figure 3H*). Next, we used STD-MO, tp53-MO, and Δ113-tp53-MO data at 48 hpf for statistical analysis to compare inhibition levels of retinal apoptosis between STD-MO, tp53-MO, and Δ113-tp53-MO (*Figure 4D*). Thus, STD-MO and tp53-MO data at 48 hpf are shared with these two statistical analyses. All the MO injections were done using embryos produced by *banp* heterozygous embryos, which were siblings outcrossed from the same *banp* heterozygous parent, in order to make genetic variation minimum. All the MOs were used at the same concentration.

### RT-PCR

Total RNA was extracted from embryo heads at the indicated developmental stages using a Sepasol RNA I Super (Nacalai, Japan). cDNA synthesis was carried out from 500 ng of RNA using Toyobo ReverTra Ace qPCR RT master mix with gDNA remover, according to the manufacturer's instructions. Expression profiles of the genes listed below were analyzed with the specified primers using Luna Universal qPCR Master Mix (NEB). Quantitative RT-PCR was performed using StepOnePlus (Applied Biosystems). Relative expression was calculated using the ΔΔCt method with the reference gene *ef1α*. Primers used in qRT-PCR analysis; *mdm2* (*Wilkins et al., 2013*), *p21(cdkn1a)* (*Stiff et al., 2016*), *FL tp53* and *Δ113 tp53* (*Chen et al., 2016*), *ef1α* (*McCurley and Callard, 2008*). All the primer sequences of *ccng1*, *mdm2*, *p21 (cdkn1a)*, *FL tp53*, *Δ113 tp53*, *puma (bbc3)*, *cenpt*, *ncapg*, *atm*, *atr*, *wrnip1*, and *ef1α* are provided in Key Resources Table.

### Western blotting

In total, 21 embryos were divided into 3 groups (each n=7) for 48 hpf wild-type and *banp*^rw337^ mutants, respectively. Heads were dissected in ice-cold E3 medium. Seven heads removed from each group were homogenized together in sample buffer (125 mM NaCl, 50 mM Tris-HCl pH 7.4, 0.5 mM EDTA, 1% Triton X-100, 1 X Protease inhibitor) and used as one sample. Appropriate volumes of samples were diluted to provide equal protein amounts and used for SDS-PAGE (BIO-RAD, Mini-PROTEAN TGX Gels). Western blot analysis was performed on 3 samples for wild-type and *banp*^rw337^ mutants, respectively, according to standard protocols using anti-tp53 antibody (GeneTex, GTX128135) at a 1:1000 dilution. β-actin protein level was evaluated by western blot analysis of rehybridized membranes with anti-β-actin antibody (Sigma, A5441) at a 1:5,000 dilution, and used to normalize tp53 protein level for each sample. For secondary antibodies, horseradish peroxidase-conjugated anti-rabbit or anti-mouse IgG antibodies (Amersham, NA931 and NA934) were used. Enzyme activity was detected using Immunostar LD (Wako) and luminescence was imaged and quantified using iBright 1,500 (Thermo Fisher Scientific).

### Live imaging

Banp-MO was injected into one-cell-stage eggs of *Tg[h2afv:GFP; EF1α: mCherry-zGem]* transgenic line. Banp-5misMO was used as a control. Two-dpf embryos were anesthetized using 5% (v/v) Tricane (3-amino benzoic acid-ethylester) and mounted laterally using 1% low-melting agarose on a depression slide to image retinas without left or right preference. Time-lapse images were obtained every 1.1 min using a confocal laser scanning microscope (LSM710, Zeiss) with a 40 × 0.8/1.0 W Plane Apochromat lens. Data were evaluated using IMARIS software (ver.9.1.2 Bitplane) to calculate the time taken to complete each cell division.

### RNA sequencing

Embryo heads were homogenized using a Polytron 1,200E homogenizer (KINEMATICA) with TRIzol reagent (Invitrogen). RNA was isolated using a Direct-zol RNA Miniprep Kit (ZYMO RESEARCH) from three replicates of *banp*^rw337^ mutant and wild-type embryos at 48 hpf. Each sample contained a pool of

7 embryo heads. RNA isolation was followed by library preparation for sequencing using an NEBNext Ultra II Directional RNA Library Prep Kit for Illumina according to the manufacturer's instructions. 151-base paired-end sequencing was performed using a NovaSeq6000 SP. The quality of sequenced reads was evaluated with FastQC (*Andrews, 2019*) and adaptor trimming and quality filtering were done with fastp (*Chen et al., 2018*). Cleaned reads were then mapped to the zebrafish reference genome (GRCz11) using hisat2.1.0 (*Kim et al., 2019*) and mapped reads were counted with FeatureCounts from the Subread package (*Liao et al., 2014*) using annotation from Ensembl (Danio_rerio.GRCz11.95. gtf). Differentially expressed gene (DEG) analysis was performed using the EdgeR package (*Robinson and Oshlack, 2010*). Genes with false discovery rates (FDR)<0.01 were considered significant DEGs. The EnhancedVolcano package (*Blighe et al., 2022*) was used to create a volcano plot with $Log_2$Fold-Change (wildtype – mutant) values and FDR calculated with EdgeR. Differences in alternative splicing between mutant and wild type were tested using the DEXseq package (*Anders et al., 2012*). Adjusted p-values < 0.01 were considered as significant.

## ATAC sequencing

Forty-eight-hpf embryos were used for ATAC sequencing. We used the Kaestner Lab ATAC seq protocol (*Ackermann, 2019*) until library preparation, followed by sequencing using a NovaSeq6000 SP. We generated reads for three independent *banp^{rw337}* mutant samples, as well as three independent *banp^{rw337}* wild-type sibling samples. Each sample contained a pool of 3 embryo heads. A quality check of reads was performed using FastQC. Reads from each sample were adaptor trimmed and aligned using fastp and Bowtie2 to the zebrafish reference genome (GRCz11) (*Langmead and Salzberg, 2012*) respectively. For quality control, aligned reads were filtered for low-quality reads, non-unique alignments, and PCR duplicates using samtools (*Li et al., 2009*; *Yan et al., 2020*). Moreover, alignment files were peak-called using Genrich, with removal of mitochondrial reads (*Gaspar, 2019*). Differences in differentially enriched peaks in wild-type and mutant embryos were identified using the DiffBind R package (*Stark and Brown, 2011*). The obtained peaks were annotated using the annotatePeaks tool of HOMER software (*Heinz et al., 2010*). The findMotifsGenome tool of HOMER was used to find enriched motifs in differentially expressed peaks. To compare coverage tracks across genes of interest in *banp^{rw337}* mutants and wild-type fish in RNA-seq and ATAC-seq data, distributions of mapped reads in each replicate were averaged and plotted with SparK (*Kurtenbach and William Harbour, 2019*).

## Statistics

Significant differences between groups (wild type vs mutant) were tested using GraphPad Prism (GraphPad Software Version 9.1.2). The number of samples used per experiment is presented in figures or figure legends. Tests used to calculate statistical significance are mentioned in figure legends. All data are presented as means ± SDs. A p value of 0.05 was considered statistically significant. In the histogram, p*<0.05, p**<0.01, p***<0.001, p****<0.0001, and ns (not significant) are indicated.

## Acknowledgements

We thank four former lab members, Masahiro Yamaguchi, Noriko Tonou-Fujimori, Yukihiro Yoshimura, Eri Oguri for supporting the initial phase of mapping and cloning experiments, and three technical staff members, Jeff Liner, Mamoru Fujiwara, and Tetsuya Harakuni for supporting mutant fish maintenance, immunohistochemistry, and DNA construction. We thank SQC, Research support division, Okinawa Institute of Science and Technology Graduate University for assistance in sequencing experiments. We thank Steven D Aird for editing the manuscript. Funding This work was supported by a grant from the Okinawa Institute of Science and Technology Graduate University to IM.

## Additional information

### Funding

| Funder | Grant reference number | Author |
| --- | --- | --- |
| Okinawa Institute of Science and Technology Graduate University | | Ichiro Masai |

The funders had no role in study design, data collection and interpretation, or the decision to submit the work for publication.

### Author contributions

Swathy Babu, Conceptualization, Resources, Data curation, Software, Formal analysis, Validation, Investigation, Visualization, Methodology, Writing - original draft, Project administration, Writing - review and editing; Yuki Takeuchi, Conceptualization, Resources, Data curation, Software, Formal analysis, Validation, Investigation, Methodology, Writing - review and editing; Ichiro Masai, Conceptualization, Resources, Data curation, Software, Formal analysis, Supervision, Funding acquisition, Validation, Investigation, Visualization, Methodology, Writing - original draft, Project administration, Writing - review and editing

### Author ORCIDs

Swathy Babu (ID) http://orcid.org/0000-0002-0612-9484
Yuki Takeuchi (ID) http://orcid.org/0000-0002-8480-5206
Ichiro Masai (ID) http://orcid.org/0000-0002-6626-6595

### Ethics

Zebrafish (Danio rerio) were maintained on a 14:10 hour light: dark cycle at 28°C. Collected embryos were cultured in E3 embryo medium (5 mM NaCl, 0.17 mM KCl, 0.33 mM $CaCl_2$, 0.33 mM $MgSO_4$) containing 0.003% 1-phenyl-2-thiouera (PTU) to prevent pigmentation and 0.01% methylene blue to prevent fungal growth. All experiments were performed on zebrafish embryos between 36 hpf and 4 dpf prior to sexual differentiation. Therefore, sexes of the embryos could not be determined. All zebrafish experiments were carried out in accordance with the Okinawa Institute of Science and Technology Graduate School's Animal Care and Use Program, which is based on the National Research Council of the National Academy's Guide for the Care and Use of Laboratory Animals and has been accredited by the Association for Assessment and Accreditation of Laboratory Animal Care (AAALAC International). The OIST Institutional Animal Care and Use Committee approved all experimental protocols (Protocol no. 2019-267, 2019-268, 2019-269, 2019-270).

### Decision letter and Author response

Decision letter https://doi.org/10.7554/eLife.74611.sa1
Author response https://doi.org/10.7554/eLife.74611.sa2

## Additional files

### Supplementary files

• Transparent reporting form
• Supplementary file 1. List of genes containing Banp motif.

### Data availability

Raw RNA-seq and ATAC-seq datasets of banprw337 mutant and wild-type siblings are available at DDBJ Sequence Read Archive (DRA012572).

The following dataset was generated:

| Author(s) | Year | Dataset title | Dataset URL | Database and Identifier |
|---|---|---|---|---|
| Babu S, Takeuchi Y, Masai I | 2022 | Comparison of expression profile between banp mutant and wildtype sibling | https://ddbj.nig.ac.jp/resource/sra-submission/DRA012572 | DDBJ, DRA012572 |

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

# Appendix 1

## Appendix 1—key resources table

| Reagent type (species) or resource | Designation | Source or reference | Identifiers | Additional information |
|---|---|---|---|---|
| strain, strain background (*Danio rerio*) | Okinawa wild type | PMID:28196805 | NA | |
| strain, strain background (*Danio rerio*) | RIKEN wild type | PMID:12702661 | ZFIN: ZDB-GENO-070802–4 | https://shigen.nig.ac.jp/zebra/ |
| genetic reagent (*Danio rerio*) | *banp*$^{rw337}$ | This paper | NA | NA |
| genetic reagent (*Danio rerio*) | *banp*$^{sa12976}$ | ZIRC | ZFIN: ZDB-ALT-130411–2647 | NA |
| genetic reagent (*Danio rerio*) | *Tg[EF1α:mCherry-zGem]*$^{oki011}$ | PMID:25260917 | ZFIN: ZDB-ALT-150128–2 | NA |
| genetic reagent (*Danio rerio*) | *Tg[EF1α:mCherry-CAAX]*$^{oki049}$ | PMID:28196805 | ZFIN: ZDB-TGCONSTRCT-181026–1 | NA |
| genetic reagent (*Danio rerio*) | *Tg[h2afv:GFP]*$^{kca6/kca66}$ | PMID:11819118 | ZFIN: ZDB-ALT-020918–4, ZDB-ALT-020717–2 | NA |
| genetic reagent (Danio) | *Tg[ath5:EGFP]*$^{rw021}$ | PMID:15728672 | ZFIN: ZDB-ALT-050627–2 | NA |
| antibody | anti-Phospho Histone H3 (Ser10) (Mouse monoclonal) | Cell signaling technology | 6G3, 9,706 | (1:500) IHC |
| antibody | pH3 (Rabbit polyclonal) | Sigma-Aldrich (Merk) | 06–570 | (1:500) IHC |
| antibody | anti-Caspase3 (Rabbit monoclonal) | BD Pharmingen | Clone C92-605 | IHC (1:200) |
| antibody | anti-Pax6 (Rabbit polyclonal) | Covance | PRB-278P | IHC (1:500) |
| antibody | zpr1 (Mouse monoclonal) | ZIRC, Eugene, Oregon | ZFIN: ZDB-ATB-081002–43 | IHC (1:100) |
| antibody | anti-glutamine synthetase (Mouse monoclonal) | Millipore | MAB302, clone GS-6 | IHC (1:100) |
| antibody | anti-PCNA (Mouse monoclonal) | Sigma-Aldrich | Clone PC10,P8825 | IHC (1:200) |
| antibody | anti-Prox1 (Rabbit polyclonal) | Gene Tex | GTX128354 | IHC (1:500) |
| antibody | anti-α-tubulin (Mouse monoclonal) | Sigma-Aldrich | T5168, clone B512 | IHC (1:1000) |
| antibody | anti-γ-H2AX (Rabbit polyclonal) | Gene Tex | GTX127342 | IHC (1:500) |
| antibody | anti-BrdU (Rat monoclonal) | abcam | Ab6326 | IHC (1:200) |
| antibody | anti-HuC/D (Mouse monoclonal) | Thermo Fisher | A-21271 | IHC (1:200) |
| antibody | anti-β-actin (Mouse monoclonal) | Sigma-Aldrich | A5441 | WB (1:5000) |
| antibody | anti-tp53 (Rabbit polyclonal) | Gene Tex | GTX128135 | WB (1:1000) |
| antibody | anti-Mouse IgG, HRP-Linked Whole Ab (Sheep polyclonal) | Cyvita | NA931 | WB (1:5000) |
| antibody | anti-Rabbit IgG, HRP-Linked Whole Ab (Donkey polyclonal) | Cyvita | NA934 | WB (1:5000) |
| recombinant DNA reagent | pBluescript II SK(+) (plasmid) | Stratagene/Agilent Technologies | NA | in vitro transcription (In situ hybridization probe synthesis) |
| recombinant DNA reagent | pCS2 (plasmid) | PMID:7926732 | NA | in vitro transcription (Capped mRNA synthesis) |
| sequence-based reagent | Primes for polymorphic marker 70,702B | This paper | PCR primers | forward: 5′-ACTTCTTATCAGGGCTGTGC-3′ reverse: 5′-TCAGTCAAGAGCAGTGAGAG-3′ |
| sequence-based reagent | Primes for polymorphic marker zC93F2D | This paper | PCR primers | forward: 5′-TGGGATCTCTTTAAGTGAGTGAG-3′ reverse: 5′-TCCAACTATGTGGGTCAAACC-3′ |

*Appendix 1 Continued on next page*

*Appendix 1 Continued*

| Reagent type (species) or resource | Designation | Source or reference | Identifiers | Additional information |
|---|---|---|---|---|
| sequence-based reagent | banp MO | This paper | Morpholino antisense oligos | 5'-CCACTAAATCTTGCTCTGACATCAT-3' |
| sequence-based reagent | banp 5misMO | This paper | Morpholino antisense oligos | 5'-CCtCaAAATgTTcCTCTcACATCAT-3' |
| sequence-based reagent | tp53 MO | PMID:12477391 | Morpholino antisense oligos | 5'-GCGCCATTGCTTTGCAAGAATTG-3' |
| sequence-based reagent | Δ113 tp53 MO | PMID:19204115 | Morpholino antisense oligos | 5'-GCAAGTTTTTGCCAGCTGACAGAAG-3' |
| sequence-based reagent | STD MO | PMID:30322969 | Morpholino antisense oligos | 5'-CCTCTTACCTCAGTTACAATTTATA-3' |
| sequence-based reagent | ccng1 | This paper | Primers for qRT-PCR | Forward primer: 5'-ccctggagattgaggatcag-3' Reverse primer: 5'cacacaaaccaggtctccaa-3' |
| sequence-based reagent | mdm2 | PMID:24147052 | Primers for qRT-PCR | Forward primer: 5'-caggaggaggagaagcagtg-3' Reverse primer: 5'-agggaaaagctgtccgactt-3' |
| sequence-based reagent | p21(cdkn1a) | PMID:26908596 | Primers for qRT-PCR | Forward primer: 5'-aagcgcaaacagaccaacat-3' Reverse primer: 5'-tcagctactggccggattt-3' |
| sequence-based reagent | FL tp53 | PMID:27539857 | Primers for qRT-PCR | Forward primer: 5'-tggagaggaggtcggcaaaatcaa-3' Reverse primer: 5'-gactgcgggaacctgagcctaaat-3' |
| sequence-based reagent | Δ113 tp53 | PMID:27539857 | Primers for qRT-PCR | Forward primer: 5'-atatcctggcgaacatttggaggg-3' Reverse primer: 5'-cctcctggtcttgtaatgtcac-3' |
| sequence-based reagent | Puma (bbc3) | This paper | Primers for qRT-PCR | Forward primer: 5'-ctgaggaggaccccacact-3' Reverse primer: 5'-tctccagttctgccagtgc-3' |
| sequence-based reagent | cenpt | This paper | Primers for qRT-PCR | Forward primer: 5'-tcatgaggagattgtggaagatg-3' Reverse primer: 5'-ggtgagctctgcgagttatt-3' |
| sequence-based reagent | ncapg | This paper | Primers for qRT-PCR | Forward primer: 5'-ctgatgtgagggagcctattt-3' Reverse primer: 5'-gagtctgtttggcctccatta-3' |
| sequence-based reagent | atm | This paper | Primers for qRT-PCR | Forward primer: 5'-cctcaaggctgtggagaact-3' Reverse primer: 5'-aggggattttctttacaccactc-3' |
| sequence-based reagent | atr | This paper | Primers for qRT-PCR | Forward primer: 5'-aggaacccaatctgccagt-3' Reverse primer: 5'-gatgtccagtgccagctctc-3' |
| sequence-based reagent | wrnip1 | This paper | Primers for qRT-PCR | Forward primer: 5'-gtgatgtgcgagaggtgataa –3' Reverse primer: 5'-acgtgtcctgctgtgattt-3' |
| sequence-based reagent | ef1α | PMID:19014500 | Primers for qRT-PCR | Forward primer: 5'-cttctcaggctgactgtgc-3' Reverse primer: 5'-ccgctagcattaccctcc-3' |
| sequence-based reagent | Genotyping primers for banp^ca12976 | This paper | Primers for sequencing | Forward primer: 5'-TGTTGATATCCATCAGTCAG-3' Reverse primer: 5'-GGTGTATAAATCACATGACC-3' |
| sequence-based reagent | Genotyping primers for banp^rw337 | This paper | PCR Primers | forward: 5'-CGATGTTGATATCCATCAGTCAGGCGATC-3'; reverse primer: 5'-GGTGCTGGTGTATAAATCACATGACCTATGGTCCTCTT-3'. |
| sequence-based reagent | Subcloning of banp full length cDNA for in Situ hybridization RNA probe synthesis | This paper | PCR Primers | Forward: 5'- cgaattcatgatgtcagagcaagatttag –3' Reverse: 5'- gctcgagtcaagtgcctggcatctggatc g-3' |
| sequence-based reagent | Subcloning of cenpt cDNA fragment for in Situ hybridization RNA probe synthesis | This paper | PCR Primers | Forward: 5'-ctggctcaaagagtgggctga-3' Reverse: 5'-agacgtcactggccaccttg-3' |

*Appendix 1 Continued*

| Reagent type (species) or resource | Designation | Source or reference | Identifiers | Additional information |
|---|---|---|---|---|
| sequence-based reagent | Subcloning of ncapg cDNA fragment for in Situ hybridization RNA probe synthesis | This paper | PCR Primers | Forward: 5'-gtcaaggaacagcgtatagag-3' Reverse: 5'-ggaaccatgatctccgattag-3' |
| sequence-based reagent | Subcloning of wrnip1 cDNA fragment for in Situ hybridization RNA probe synthesis | This paper | PCR Primers | Forward: 5'-aactgatcggagaacaaactc-3' Reverse: 5'-gcacactgggctagaataac-3' |
| commercial assay or kit | DIG RNA Labelling Kit | Roche | 11175025910 | In situ hybridization |
| commercial assay or kit | mMESSAGE mMACHINE SP6 Transcription Kit | Invitrogen | AM1340 | Capped mRNA synthesis |
| commercial assay or kit | In Situ Cell Death Detection Kit, TMR red | Roche | 12156792910 | apoptosis |
| commercial assay or kit | ReverTra Ace aPCR master mix with gDNA remover | Toyobo | FSQ-301 | cDNA synthesis |
| commercial assay or kit | Luna Universal qPCR Master Mix | NEB | M3003L | qRT-PCR |
| commercial assay or kit | Direct-zol RNA Miniprep Kit | ZYMO RESEARCH | R2050 | RNA-sequencing |
| commercial assay or kit | NEBNext Ultra II Directional RNA Library Prep Kit | NEB | E7760 | Illumina RNA-sequencing |
| chemical compound, drug | Acridine orange | WALDECK (CHROMA) | 1B-307 | Live cell death detection |
| software, algorithm | IMARIS | Bitplane | ver.9.1.2 | http://www.bitplane.com/imaris; RRID: SCR_007370 |
| software, algorithm | Image J | NIH | Version 2.1.0/1.53 c | Percentage area calculation |
| software, algorithm | ZEN 2012 | Zeiss | LSM710 (Version: 14.0.25.201) | Percentage area calculation |
| Software, algorithm | GraphPad Prism | GraphPad Software | Version 9.1.2 | https://www.graphpad.com/scientific-software/prism/ |
| other | SYTOX Green | Molecular Probes | S34862 | IHC (1:1000) |
| other | rhodamine-conjugated phalloidin | Molecular Probes | R415 | Filamentous actin (F-actin) stain IHC (1: 40) |
| other | Restore PLUS Western Blot Stripping Buffer | Thermo Scientific | 46,430 | Remove high-affinity antibodies from membranes (western blot) |

