## [Editor Report]

A considerable knowledge gap exists in the understanding of BANP gene roles in vivo due to its essentiality, causing embryonic lethality in mice and humans. The authors here address this by exploiting the retinal cell proliferation and neurogenesis in zebrafish and define Banp roles in replication stress responses and mitosis involving p53, ATM and ATR signaling.

---

## [Decision Letter]

**Decision letter after peer review:**

Thank you for submitting your article "Banp regulates DNA damage response and chromosome segregation during the cell cycle in zebrafish retina" for consideration by *eLife*. Your article has been reviewed by 3 peer reviewers, and the evaluation has been overseen by a Reviewing Editor and Richard White as the Senior Editor. The following individual involved in review of your submission has agreed to reveal their identity: Sarah R Hengel (Reviewer #1).

Essential revisions:

1) Substantially revise the text. All reviewers agreed that the data could be of potential interest and impact, but raised concerns about the presentation of the results. As such the impact and relevance beyond the biologist in the field of zebrafish eye development was difficult to discern. It needs to be written so that someone outside of the retina field understand the implications.

2) Add additional experimentation to better illustrate the new Banp functions in replication stress and mitosis. As is the reviewers agreed that in particular the proposed function during mitosis appears internally inconsistent and contradictory. As one example, the term mitotic arrest usually reflects a failure to complete mitosis, typically as a metaphase arrest. Once anaphase is committed to (chromosome segregation), there is no arrest that the reviewers are aware of e.g. in late anaphase.

*Reviewer #2 (Recommendations for the authors):*

1. Banp mutants have numerous defects, and perhaps this is not unexpected for a nuclear matrix protein. I'm left wondering what insights are gained from the study beyond that the nuclear matrix is required for numerous cell cycle events?

2. Why did the authors focus on the eye? It is unclear whether this study revealed a sensitivity to eye development regarding nuclear matrix function specifically, or it was just a convenient place in the animal to look.

3. I found the conclusions regarding mitosis to be contradictory. The authors at first emphasize mitotic arrest, but then characterize chromosome segregation defects. How can chromosomes segregate if cells are arrested in mitosis?

4. It would be important to know whether the authors can rule out that S-phase defects cause the M phase defects, or vice versa. Could there be a primary defect, rather than multiple independent defects as the authors conclude?

*Reviewer #3 (Recommendations for the authors):*

To increase the impact of the work, provide a better understanding of how loss of banp leads to induction of DNA damage (currently presented as a question mark in Figure 7).

Reorganize the figures, the text or both so that the reader can easily recognize all the data that support the same point.

Make sure that all conclusions are supported by data, either new or published.

Line 229: 'Histone H3 (pH3) antibody, which labels mitotic cells in pro-metaphase…' Actually, pH3 is seen also in other stages of mitosis and not just pro-metaphase, so the statement should be corrected.

Line 375: If 'mitotic arrest in banprw337 mutants is mostly mediated by a tp53-independent mechanism.' Provide an explanation for the reduction in pH3 cells with FL tp53 MO in Figure 4E.

Line 319: why do the authors suggest that 'FL tp53 triggers Δ113tp53 transcription in banprw337 mutants'? Is full length p53 known to promote its own transcription from an alternate promoter? If so, please provide data or a citation.

[Editors' note: further revisions were suggested prior to acceptance, as described below.]

Thank you for resubmitting your work entitled "Banp regulates DNA damage response and chromosome segregation during the cell cycle in zebrafish retina" for further consideration by *eLife*. Your revised article has been evaluated by Richard White (Senior Editor), Katharina Schlacher (Reviewing Editor) and reviewers.

The reviewers noted that the manuscript has been improved but there are some remaining issues that need to be addressed before publication, as outlined below:

1) Line 288-289: The authors changed 'prometaphase' to 'metaphase', but this is inaccurate. The correct statement should indicate that pH3 is seen at many stages of mitosis including prometaphase, metaphase and early anaphase.

2) A major concern from the previous version remains to be clarified better, which is the relationship between the DNA damage in S-phase and the mitotic defects. In particular it needs to be stated clearly if these two defects are truly independent or if one precedes the other. If the S phase defects are independent of the mitotic defects, then few PH3+ cells will have foci. The authors state line 419-425 "There is no significant difference in the pH3+ cell fraction between in the total retinal cells and γ-H2AX+ retinal cells (Figure 4—figure supplement 1J). While there is no significant difference it is slightly decreased so this data could potentially address this issue, but should be clearly presented and interpreted. E.g. if S-phase defects are independent of M-phase defects but S-phase defects precede mitosis, then what happens to those damaged S-phase cells? A concise clarification in the text can address this issue.

3) The cartoons/models should be revised into multiple model panels that are each simplified and concentrate on the essential information, as well as a temporal progression be added (e.g. currently it appears that the BrdU^+^ cells are derived from M phase cells, is this correct?

An additional suggestion but not a requirement is to consider adding wrnip1 data to the abstract. The link to DNA damage is one of the key findings of the work.

---

## [Author Response]

Essential revisions:1) Substantially revise the text. All reviewers agreed that the data could be of potential interest and impact, but raised concerns about the presentation of the results. As such the impact and relevance beyond the biologist in the field of zebrafish eye development was difficult to discern. It needs to be written so that someone outside of the retina field understand the implications.

To emphasize the impact of our work beyond zebrafish eye development, we have revised the text as follows.

1) In the introduction, we have moved the explanation of Banp protein to the first paragraph. Then we introduce the zebrafish retina and its contribution to research on cell-cycle regulation and DNA damage response. We also trimmed some details from the description of mechanisms of retinal development. We believe that these arrangements better emphasize the impact of our findings on Banp and will enable readers to more easily understand the research background of our studies about the role of Banp in cell-cycle and DNA damage response.

2) In the results, we have combined the paragraph on reduced expression of chromosome segregation regulators *ncapg* and *cenpt* in *banp* mutants with the paragraph on live imaging of chromosome segregation phenotypes in banp morphants. These data are shown in a new Figure 5. Accordingly, the impact of Banp motifs on Banp downstream target genes is summarized in an independent section and shown in a new Figure 6. These rearrangements make the role of Banp in cell-cycle regulation more clear, and should increase impact of our findings.

To address the concerns above, we have revised the manuscript as follows.

1) To better illustrate the new Banp functions in replication stress and DNA damage repair, among potential direct targets of Banp, we examined *wrnip1*, which reportedly protects stalled forks from degradation and promotes fork restart after replication stress. We show that loss of Banp compromise expression of *wrnip1,* potentially leading to one cause of replication stress (Figure 6 and Figure 6—figure supplement 2).

2) For data consistency, we have changed “mitotic arrest” into “mitotic cell accumulation” or simply “mitotic defects” in descriptions of *banp* mutant phenotypes, because “mitotic arrest” may mislead the readers to think that M phase progression is permanently arrested during mitosis, which is contradictory to our live imaging observations that mitosis is markedly prolonged, but eventually completed in *banp* morphants.

We have provided more detail about these additional experiments in a point-by-point response to each referee’s comments. Line number in this letter indicates the position of our corrections in the revised manuscript. Please access the line number in the manuscript with the option “Show all Revisions inline” in Microsoft word. The line number may change if track changes are partially or completely masked.

2) Add additional experimentation to better illustrate the new Banp functions in replication stress and mitosis. As is the reviewers agreed that in particular the proposed function during mitosis appears internally inconsistent and contradictory. As one example, the term mitotic arrest usually reflects a failure to complete mitosis, typically as a metaphase arrest. Once anaphase is committed to (chromosome segregation), there is no arrest that the reviewers are aware of e.g. in late anaphase.Reviewer #2 (Recommendations for the authors):1. Banp mutants have numerous defects, and perhaps this is not unexpected for a nuclear matrix protein. I'm left wondering what insights are gained from the study beyond that the nuclear matrix is required for numerous cell cycle events?

As we mentioned in the Introduction, BANP was originally identified as a nuclear protein that binds matrix-associated regions (MARs). MARs are regulatory DNA sequences mostly present upstream of various promoters. MAR-binding proteins interact with numerous chromatin-modifying factors and regulate gene transcription. In addition, it was reported that BANP suppresses tumor growth, and that loss of BANP heterozygosity is associated with several cancers in humans. So, before we started this *banp* mutant analysis, we expected that loss of Banp might cause defects in the cell cycle. However, because the majority of prior studies on BANP have been done using in vitro systems, its physiological function was still ambiguous. Very recently, it was reported that BANP functions as a transcription factor that binds to Banp motifs and regulates essential metabolic genes. In this study, rather than focusing on the MAR domain, we used this Banp motif to search for direct transcriptional targets of Banp that may function in cell proliferation and differentiation in zebrafish retina. Our study provides the first in vivo evidence that Banp serves as an essential transcription activator of cell cycle genes, including *cenpt*, *ncapg*, and *wrnip1* via Banp motifs. We believe that such a list of Banp direct target genes provides a new research avenue to discover more precisely how Banp functions in tumor suppression and that it will contribute to medical research on cancer therapy.

Our study did not investigate how the nuclear matrix itself is involved in Banp mutant phenotypes. However, since it is likely that the interaction between MAR domains and nuclear matrix may influence chromatin organization in the nucleus, BANP functions must depend on nuclear matrix configuration. So, while this question is interesting, we think it is beyond the scope of our current study. In addition, we are afraid that the term “matrix-associated nuclear protein” might mislead people to think that Banp is a regulator of nuclear matrix. To better clarify the relationship between Banp and nuclear matrix, we have revised “nuclear matrix-associated protein” -> “nuclear matrix associated region-binding protein” in the text.

2. Why did the authors focus on the eye? It is unclear whether this study revealed a sensitivity to eye development regarding nuclear matrix function specifically, or it was just a convenient place in the animal to look.

Historically, molecular and cellular mechanisms that regulate cell proliferation and differentiation in the nervous system has been intensively studied using the vertebrate retina, because retinal neuronal cell types are fewer than those of other brain regions and its neural circuits are also simpler than those of other brain regions. Furthermore, many research groups, including us, have identified zebrafish retinal mutants, including mutants that show defects in cell-cycle regulation and DNA damage response. Indeed, our group has investigated this topic using retinal apoptotic mutants for the last 20 years. Thus, we focus on the zebrafish retina, because the retina is an excellent in vivo model system to dissect mechanisms of cell-cycle regulation and DNA damage response.

To emphasize the importance of this excellent in vivo model system to researchers beyond the retinal community, we have revised in the Introduction as follows.

The developing retina is a highly proliferating tissue, in which a spatiotemporal pattern of neurogenesis is tightly coordinated by cell-cycle regulation. So, vertebrate retina provides a great model for studying how cell-cycle regulation, including DNA damage response ensures neurogenesis and subsequent cell differentiation. (Line 109-113).

3. I found the conclusions regarding mitosis to be contradictory. The authors at first emphasize mitotic arrest, but then characterize chromosome segregation defects. How can chromosomes segregate if cells are arrested in mitosis?

We apologize for the confusion due to our incorrect usage of the term “mitotic arrest.” Mitotic arrest was one of possibilities that we considered when first examining *banp* mutant phenotypes, in which we just observed accumulation of mitotic (pH3+) cells. However, when we examined mitosis in Banp morphants using live imaging, we found that mitosis duration is significantly prolonged because of chromosome segregation defects in Banp morphants, but that all 28 mitoses we examined eventually completed cytokinesis. Thus, we finally concluded that mitotic cells are not permanently arrested in M phase, but that mitosis is prolonged. To prevent confusion, we have changed “mitotic arrest” to “mitotic cell accumulation” or simply “mitotic defects” in the Results section on *banp* mutant phenotype analysis (shown in Figures 2 and 4).

4. It would be important to know whether the authors can rule out that S-phase defects cause the M phase defects, or vice versa. Could there be a primary defect, rather than multiple independent defects as the authors conclude?

We thank reviewer #3 for this suggestion. Interdependence between S phase defects and M phase defects is important to correctly interpret the data on cell-cycle regulation, especially cell-cycle checkpoint and DNA damage response. Indeed, there are interesting reports using in vitro cell culture systems indicating that replication stress induces mitotic death, through specific pathways for example, Masamsetti et al., 2019, Nat. Comm. 10.4224. However, this topic is still challenging to dissect in vivo.

In terms of our findings on Banp functions in zebrafish, we found that two chromosome segregation regulators, *ncapg* and *cenpt*, are direct transcription targets of Banp, and that it is likely that loss of Banp causes mitotic defects through downregulation of *cenpt* and *ncapg*. From this point, we conclude that mitotic defects are primary effects of the loss of Banp. The next question is how the loss of Banp stalls DNA replication forks and causes subsequent cell death. To address this question, we examined whether Banp direct targets include cell-cycle regulators, especially in S phase. We found that *wrnip1* is an interesting candidate, because Wrnip1 reportedly protects stalled replication forks and promotes fork restart after DNA replication stress. In addition, Wrnip1 functions in interstrand crosslink repair (ICR). We found that the mRNA expression level of *wrnip1* is markedly decreased in *banp* mutants, suggesting the possibility that DNA replication stress may be caused by reduction of *wrnip1* expression in *banp* mutants. We present these data in new Figure 6—figure supplement 2. We have revised the possible role of Banp in cell-cycle regulation in new Figure7. Under this scenario, we consider it likely that loss of Banp may cause DNA replication stress through downregulation of S phase regulators, independent of mitotic defects.

However, we cannot exclude the possibility that DNA replication stress causes mitotic defects in *banp* mutants. Masamsetti et al., 2019, Nat. Comm. 10.4224. revealed that replication stress induces spindle assembly checkpoint (SAC)-dependent mitotic arrest and subsequent mitotic death when tp53 activity is inhibited. We showed that cell death in zebrafish *banp* mutant retinas was fully suppressed by tp53-MO at 48 hpf, but still occurred at 72 hpf, although there was no significant difference between wildtype and *banp* mutants (Figure 3GH). In the manuscript, we mentioned the possibility that some tp53-independent mechanism induces retinal apoptosis in *banp* mutants after 48 hpf. An alternative possibility is that most cell death in *banp* mutants depends on tp53; however, replication stress persisting in *banp* mutants injected with MO-tp53 may cause SAC-mediated mitotic death, as reported by Masamsetti et al., 2019. Future studies will be necessary to clarify this possibility.

Reviewer #3 (Recommendations for the authors):To increase the impact of the work, provide a better understanding of how loss of banp leads to induction of DNA damage (currently presented as a question mark in Figure 7).

Please see our response to Essential revision #2 and Reviewer 2 point #3. We found that *wrnip1* mRNA expression is drastically reduced in *banp* mutants, which may cause DNA replication stalling and abnormal phenotypes.

Reorganize the figures, the text or both so that the reader can easily recognize all the data that support the same point.Make sure that all conclusions are supported by data, either new or published.

A similar concern was raised by the editors (Essential revision #1). Please see our response.

Line 229: 'Histone H3 (pH3) antibody, which labels mitotic cells in pro-metaphase…' Actually, pH3 is seen also in other stages of mitosis and not just pro-metaphase, so the statement should be corrected.

We thank the reviewer for this correction. We changed “pro-metaphase” to “metaphase” (Line 282-283).

Line 375: If 'mitotic arrest in banprw337 mutants is mostly mediated by a tp53-independent mechanism.' Provide an explanation for the reduction in pH3 cells with FL tp53 MO in Figure 4E.

We are grateful for these comments. Since mitotic cell accumulation is significantly reduced by FL tp53-MO, there is a substantial contribution of FL tp53 to mitotic cell accumulation. So, we have revised the sentences (Line 438-448).

Compared with Standard-MO injection controls (n=376.6±74.8/section), the number of pH3+ cells in *banp^rw337^* mutants was significantly decreased by FL tp53-MO (n=262.4±68.3/section) (Figure 4C, 4E). However, the number of pH3+ cells was still higher in *banp^rw337^* mutant retinas injected with FL tp53-MO than in wild-type control retinas injected with FL tp53-MO, indicating that there is a substantial tp53independent fraction. Furthermore, injection of ∆113tp53-MO did not significantly rescue mitotic phenotypes in *banp^rw337^* mutants (n=303.8±46.6/section) (Figure 4C, 4F). These data suggest a tp53-independent mechanism that causes mitotic cell accumulation in *banp^rw337^* mutants.

Line 319: why do the authors suggest that 'FL tp53 triggers Δ113tp53 transcription in banprw337 mutants'? Is full length p53 known to promote its own transcription from an alternate promoter? If so, please provide data or a citation.

We have cited a paper reporting the evidence (Line 360-362). The authors claim that FL tp53 initially activates transcription of ∆113tp53 after tp53 protein is stabilized in response to cellular stress (Chen et al., 2009).

[Editors' note: further revisions were suggested prior to acceptance, as described below.]

The reviewers noted that the manuscript has been improved but there are some remaining issues that need to be addressed before publication, as outlined below:1) Line 288-289: The authors changed 'prometaphase' to 'metaphase', but this is inaccurate. The correct statement should indicate that pH3 is seen at many stages of mitosis including prometaphase, metaphase and early anaphase.

We have revised accordingly.

Line 227: mitotic cells in prometaphase, metaphase and early anaphase (Prigent and Domitrov, 2013)

2) A major concern from the previous version remains to be clarified better, which is the relationship between the DNA damage in S-phase and the mitotic defects. In particular it needs to be stated clearly if these two defects are truly independent or if one precedes the other. If the S phase defects are independent of the mitotic defects, then few PH3+ cells will have foci. The authors state line 419-425 "There is no significant difference in the pH3+ cell fraction between in the total retinal cells and γ-H2AX+ retinal cells (Figure 4—figure supplement 1J). While there is no significant difference it is slightly decreased so this data could potentially address this issue, but should be clearly presented and interpreted. E.g. if S-phase defects are independent of M-phase defects but S-phase defects precede mitosis, then what happens to those damaged S-phase cells? A concise clarification in the text can address this issue.

Thank you for your suggestions. We agree that it is important to clarify how S phase defects and M phase defects influence each other. As you suggested, it is interesting that pH3+ cell fraction in g-H2AX+ retinal cells is lower than that of total retinal cells, although there is no statistical difference. At 48 hpf, *banp* mutants show increases in

1

g-H2AX+ cells, pH3 + cells, and apoptotic cells, compared with wild-type control. The reduction of pH3+ cell fraction in g-H2AX+ retinal cells suggest that g-H2AX+ retinal cells are eliminated by apoptosis in S phase prior to mitosis. In addition, M phase defects are likely to be independent of replicative damages mostly introduced in S phase.

We have added the sentences below in the Results section (Line 364-369).

“Despite of no statistical significance, interestingly, averaged fraction of pH3+ cells in γ-H2AX+ retinal cells was lower than that of total retinal cells. Since *banp^rw337^* mutants show increase in γ-H2AX + cells, pH3 + cells and apoptotic cells at 48hpf, it is likely that DNA damaged cells are eliminated by apoptosis in S phase prior to mitosis. This also suggest that M phase defects may be independent of replicative DNA damage introduced in S phase.”

Furthermore, regarding the question on, “if S-phase defects are independent of Mphase defects but S-phase defects precede mitosis, then what happens to those damaged S-phase cells?”, it is difficult to examine S phase defects and M phase defects separately because both defects appear at 48 hpf. However, our observations indicate that mitosis in *banp* mutants is eventually completed, and that apoptosis is not overlapped with accumulated pH3+ cell area, suggesting that M phase defects are not associated with apoptosis but only prolonged mitosis duration. So, DNA damage is likely to be inherited by daughter cells through the mitosis, resulting in apoptosis in immature neurons. To clarify how S- and M-phase defects influence each other in *banp* mutants, we have added the description on the summary of *banp* mutant phenotype in the Results section (Line 502-509) in Figure 7B and its legends (Line 1470-1480), which we hope guide readers to understand how S- and M-phase defects influence each other during cell-cycle progression in *banp* mutants.

3) The cartoons/models should be revised into multiple model panels that are each simplified and concentrate on the essential information, as well as a temporal progression be added (e.g. currently it appears that the BrdU^+^ cells are derived from M phase cells, is this correct?

We consider that the cartoons/models pointed by editors above are Figures 2EF. Since the original Figures 2EF globally describe retinal phenotypes of *banp* mutants, especially only TUNEL and pH3+ cells at 53 to 77 hpf, this explanation does not clearly represent the interpretation drawn from the data of Figure 2 and Figure 2—figure supplement 1. We have revised these two figure panels to clarify what happens at a cellular level: cell-cycle progression and neurogenesis along the apico-basal axis of the neural retina as well as developmental stages. We replaced new cartoons in Figure 2EF and also revised the text of Results section (Line 277-284) and their legends (Line 1288-1311).

An additional suggestion but not a requirement is to consider adding wrnip1 data to the abstract. The link to DNA damage is one of the key findings of the work.

We have added *wrnip1* data to the abstract.